# Provable Model-Parallel Distributed Principal Component Analysis with Parallel Deflation

Fangshuo Liao[1], Wenyi Su[1], Anastasios Kyrillidis[1, 2]
[1]Computer Science Department, Rice University
[2]Ken Kennedy Institute, Rice University
{Fangshuo.Liao,bs82,anastasios}@rice.edu

We study a distributed Principal Component Analysis (PCA) framework where each worker targets a distinct eigenvector and refines its solution by updating from intermediate solutions provided by peers deemed as "superior". Drawing intuition from the deflation method in centralized eigenvalue problems, our approach breaks the sequential dependency in the deflation steps and allows asynchronous updates of workers, while incurring only a small communication cost. To our knowledge, a gap in the literature – *the theoretical underpinning of such distributed, dynamic interactions among workers* – has remained unaddressed. This paper offers a theoretical analysis explaining why, how, and when these intermediate, hierarchical updates lead to practical and provable convergence in distributed environments. Despite being a theoretical work, our prototype implementation demonstrates that such a distributed PCA algorithm converges effectively and in scalable way: through experiments, our proposed framework offers comparable performance to EigenGame-$\mu$, the state-of-the-art model-parallel PCA solver.

## 1. Introduction

Currently, datasets have gotten large, encompassing billions, if not trillions, of entries spanning various domains [1–10]. This scale advanced various distributed optimization protocols, such as federated learning [11], and, notably, the development of multiple distributed ML software packages [12, 13]. Specialized frameworks such as Ray [14], Spark [15], Hadoop [16], and JAX [17] have become popular due to their ability to speed computations significantly.

However, at the algorithmic level, most distributed implementations simulate the behavior of the centralized versions of the underlying algorithms. That is, how distributed algorithms navigate the parameter landscape is often designed such that we achieve a similar outcome as if data is available in one location. There are a few key reasons for this:

- **Mathematical Understanding**: When there is sufficient theoretical understanding of the centralized version, it is often a desired goal to attain the same result by designing algorithms to emulate the centralized counterparts. This ensures consistency and theoretical understanding.

- **Algorithm Simplicity**: Since centralized algorithms are better understood, distributed variants that replicate the algorithms' outcomes enjoy the same simplicity and interpretation.

- **Benchmarking**: By simulating the centralized execution, comparing the accuracy and convergence properties of the distributed implementation in practice becomes easier.

Yet, precisely simulating centralized algorithms in a distributed fashion could pose some challenges. Take as a characteristic feature the notion of *synchrony* in distributed implementations, as this leads to training dynamics closer to centralized training. Synchronization among workers means proper orchestration [18, 19]: Synchronized implementations that wait for some or all workers to finish each iteration before proceeding can suffer from stragglers and load imbalance [20, 21]. Yet, while asynchronous motions seem like a favorable alternative, developing an asynchronous learning method is often complicated [22–24], set aside the lack of theoretical understanding in many cases.

Second Conference on Parsimony and Learning (CPAL 2025).

This work addresses a fundamental question in distributed systems: whether and how to orchestrate workers, by focusing on a specific problem - Principal Component Analysis (PCA) [25–32]. While PCA is conceptually straightforward, developing efficient distributed implementations remains an active research challenge, recently reinvigorated by the introduction of EigenGame [33, 34]. EigenGame departs from traditional data-parallel approaches to distributed PCA, where each worker processes a portion of the data or covariance matrix. Instead, it introduces a model-parallel framework inspired by game theory, where each worker computes specific principal components. This novel approach enables PCA computation on massive datasets, as demonstrated with the Meena conversation model and ResNet-200 [35] activations on ImageNet [36]. However, EigenGame's theoretical guarantees rely on a strict hierarchical structure: each worker handles exactly one component and must wait for the convergence of higher-priority principal components (those with larger eigenvalues) before proceeding. While effective, this hierarchical dependency has theoretical limitations, as the impact of approximation errors in higher-ranked eigencomponents on subsequent calculations remains uncharacterized in [33].

**Our approach and contributions.** This work advances model-parallel distributed PCA by building upon the collaborative computation framework introduced by [33, 34]. Our approach proposes a provable parallel computation of principal components without imposing strict sequential dependencies among workers. Our primary contributions are:

- We introduce a distributed PCA framework that fundamentally transforms computational dynamics. Different from traditional sequential computational model, Our approach enables multiple workers to compute distinct principal components in parallel.

- In cases where the covariance is unknown or cannot be efficiently estimated, our algorithm can be modified to accommodate data that comes in mini-batches.

- We provide theory that validates the convergence properties of our proposed algorithm. By formalizing the interaction between parallel computations and convergence rates, we establish a theoretical benchmark for distributed PCA algorithms. This contribution underscores our algorithm's efficiency and enhances the understanding of parallel deflation processes in PCA.

- Through experiments, we demonstrate the practical efficacy of our algorithm. Our approach meets the performance of existing algorithms on datasets as large as ImageNet [36], justifying our theory and highlighting the applicability of our method.

## 1.1. Related Works

**Eigenvector-based approaches.** PCA has been fundamental to statistical data analysis since its introduction by Pearson in 1901 [cite]. The method was later formalized within a multivariate analysis framework by Hotelling [26], establishing its theoretical foundations. In its classical form, PCA involves computing an empirical covariance matrix from the data, followed by its eigendecomposition. This formulation allows the application of numerous efficient numerical methods, including QR decomposition [37], xLAHQR [38], the Lanczos method [39], and ARPACK [40], some of which are implemented in numerical linear algebra packages such as ScaLAPACK [41]. These methods are effective but often require complete knowledge of the covariance matrix prior to computation.

**Centralized stochastic approaches.** With large datasets, iterative and gradient-based methods for PCA have gained prominence. Krasulina and Oja & Karhunen proposed two of the earliest stochastic gradient descent methods for online PCA [42, 43]. The application of the least square minimization to the PCA has also received attention [44–47]. More recently, [48] and [49] have proposed efficient stochastic optimization methods that adapt to the streaming model of data (stochastic) and focus on the theoretical guarantees of gradient-based methods in such non-convex scenarios; see also [50–53]. Other approaches include manifold methods [54–57], Frank-Wolfe methods [58], Gauss-Newton methods [59], coordinate descent methods [60], accelerated methods [61], as well as variants of the PCA problem itself [62–66]. Nevertheless, these methods are primarily designed as centralized algorithms.

**Data-parallel distributed approaches.** Prior distributed PCA approaches span several key directions. One line of work utilizes randomized linear algebra and SVD projections in distributed set-

tings, yielding strong theoretical guarantees [67–70]. For distributed subspace computation, recent methods combine FedAvg with Orthogonal Procrustes Transformations [71–74]. Approaches for computing leading principal components leverage both convex [75] and Riemannian optimization for improved efficiency [76, 77]. Notable recent advances include an asynchronous Riemannian gradient method that achieves low computational and communication costs [56]. The field has also expanded to address specialized scenarios, including Byzantine-robust computation [78, 79], streaming data analysis [80, 81], shift-and-invert preconditioning [82], and coreset-based approaches [83].

**Model-parallel distributed approaches.** While most prior work focuses on data-parallel approaches, where each machine computes all principal components using local data, DeepMind's EigenGame [33] introduced a novel model-parallel framework. Their approach reformulates PCA as a collaborative game, where each principal component computation acts as a player maximizing its utility through Riemannian gradient ascent. Though initially presented as a sequential process with proved convergence guarantees, EigenGame extends to a distributed setting where components are optimized simultaneously across machines. While this parallel extension offers practical benefits, its theoretical convergence properties remain unanalyzed, a limitation that persists in subsequent improvements [34].

Our work complements existing literature mostly theoretically, but also practically. By eliminating the requirement for sequential completion of principal components, our algorithmic framework achieves comparable empirical performance to EigenGame [33] on large-scale datasets. Crucially, we establish rigorous convergence guarantees for parallel computation, providing the theoretical foundation that has been missing in existing model-parallel approaches.

## 2. Problem statement and background

Let $\mathbf{Y} \in \mathbb{R}^{n \times d}$ be the matrix representing an aggregation of $n$ properly scaled, centered data points, each with $d$ features. The empirical covariance matrix is given by $\mathbf{\Sigma} = \mathbf{Y}^\top \mathbf{Y} \in \mathbb{R}^{d \times d}$. Let $\mathbf{u}_k^\star$ and $\lambda_k^\star$ be the $k$th eigenvector and eigenvalue of $\mathbf{\Sigma}$, with $\lambda_1^\star \geq \cdots \geq \lambda_d^\star$. Then $\mathbf{u}_k^\star$ is the $k$th principal component of the data matrix $\mathbf{Y}$. Therefore, when $\mathbf{\Sigma}$ can be easily computed, principal component analysis aims at finding the top-$K$ eigenvectors of the empirical covariance matrix $\mathbf{\Sigma}$, where $K \leq d$.

**The leading eigenvector problem.** Finding the leading eigenvector is the cornerstone of finding multiple eigenvectors, and is thus utilized by many PCA algorithms. Mathematically, the problem of finding the leading eigenvector $\mathbf{u}_1^\star$ can be formulated as the following optimization problem:
$$\mathbf{u}_1^\star = \underset{\mathbf{v} \in \mathbb{R}^d : \|\mathbf{v}\|_2 = 1}{\arg\max} \ \mathbf{v}^\top \mathbf{\Sigma} \mathbf{v}. \tag{1}$$

In practice, algorithms like power iteration and Hebb's rule are used to solve the leading eigenvector.

**Definition 1** (Power Iteration). *The power iteration algorithm* PowIter $(\mathbf{\Sigma}, \mathbf{v}, T)$ *outputs a vector* $\mathbf{x}_T$ *based on the following iterates:*
$$\mathbf{x}_0 = \mathbf{v}; \quad \hat{\mathbf{x}}_{t+1} = \mathbf{\Sigma} \mathbf{x}_t; \quad \mathbf{x}_{t+1} = \hat{\mathbf{x}}_{t+1} / \|\hat{\mathbf{x}}_{t+1}\|_2.$$

**Definition 2** (Hebb's Rule). *The Hebb's Rule* Hebb $(\mathbf{\Sigma}, \mathbf{v}, T)$ *with some fixed step size* $\eta$ *outputs a vector* $\mathbf{x}_T$ *based on the following iterates:*
$$\mathbf{x}_0 = \mathbf{v}; \quad \hat{\mathbf{x}}_{t+1} = \mathbf{x}_t + \eta \mathbf{\Sigma} \mathbf{x}_t; \quad \mathbf{x}_{t+1} = \hat{\mathbf{x}}_{t+1} / \|\hat{\mathbf{x}}_{t+1}\|_2.$$

Under mild assumptions, the output $\mathbf{x}_T$ of both the power iteration and Hebb's rule converges to the top eigenvector of the input matrix $\mathbf{\Sigma}$, as the number of steps $T \to \infty$. Notably, the power iteration enjoys a linear convergence rate [49].

**Top-$K$ eigenvector using sequential deflation.** An extension of (1) is the top-$K$ eigenvector problem, where one aims to find $\mathbf{u}_1^\star, \ldots \mathbf{u}_K^\star$. Since $\mathbf{u}_1^\star, \ldots, \mathbf{u}_K^\star$ form an orthonormal set, finding the top-$K$ eigenvector can be mathematically formulated as:
$$\mathbf{U}^\star = [\mathbf{u}_1^\star, \ldots, \mathbf{u}_K^\star] \in \underset{\mathbf{V} \in \{\mathbf{Q}_{:,:K} : \mathbf{Q} \in \mathrm{SO}(d)\}}{\arg\max} \langle \mathbf{\Sigma} \mathbf{V}, \mathbf{V} \rangle, \tag{2}$$

where $\mathrm{SO}(d)$ denotes the group of rotations about a fixed point in $d$-dimensional Euclidean space. A classical way to solve (2) is through *deflation* [26]. Deflation operates in the following manner.

Once the top component $\mathbf{u}_1^\star$ is approximated, the matrix $\boldsymbol{\Sigma}$ undergoes further processing to reside in the subspace orthogonal to the one spanned by the first eigenvector. This process is iterated by finding the leading eigenvector as in (1) on the deflated matrix, resulting in an approximation of the second component $\mathbf{u}_2^\star$, and so forth, as described below:

$$\boldsymbol{\Sigma}_1 = \boldsymbol{\Sigma}; \quad \mathbf{v}_k = \mathtt{Top1}\left(\boldsymbol{\Sigma}_k, \hat{\mathbf{v}}_k, T\right);$$
$$\boldsymbol{\Sigma}_{k+1} = \boldsymbol{\Sigma}_k - \mathbf{v}_k \mathbf{v}_k^\top \boldsymbol{\Sigma}_k \mathbf{v}_k \mathbf{v}_k^\top, \tag{3}$$

where $\mathtt{Top1}\left(\boldsymbol{\Sigma}_k, \hat{\mathbf{v}}_k, T\right)$ abstractly denotes any iterative algorithm initialized at $\hat{\mathbf{v}}_k$ and returns a normalized approximation of the top eigenvector of the deflated matrix $\boldsymbol{\Sigma}_k$ after $T$ iterations of execution. Consider the eigendecomposition $\boldsymbol{\Sigma} = \sum_{k'=1}^d \lambda_{k'}^\star \mathbf{u}_{k'}^\star \mathbf{u}_{k'}^{\star\top}$. When $T \to \infty$ and $\mathtt{Top1}\left(\boldsymbol{\Sigma}_k, \hat{\mathbf{v}}_k, T\right)$ solves the top eigenvector of $\boldsymbol{\Sigma}$ exactly, one can show that $\boldsymbol{\Sigma}_k = \sum_{k'=k}^d \lambda_{k'}^\star \mathbf{u}_{k'}^\star \mathbf{u}_{k'}^{\star\top}$ and $\mathbf{v}_k = \mathbf{u}_k^\star$. However, when $T$ is finite, it is shown in [84] that each $\mathtt{Top1}\left(\boldsymbol{\Sigma}_k, \hat{\mathbf{v}}_k, T\right)$ produces a non-negligible error that accumulates and propagates through the deflation process.

**Stochastic algorithm to find top-1 principal component.** When the dataset becomes large, the covariance matrix $\boldsymbol{\Sigma}$ may not be efficiently computed, making the previous routine of first computing the covariance matrix and then its eigenvector infeasible. Alternatively, people estimate $\boldsymbol{\Sigma}$ with $\hat{\boldsymbol{\Sigma}} = \hat{\mathbf{Y}}^\top \hat{\mathbf{Y}}$, where $\hat{\mathbf{Y}}$ is a mini-batch of the dataset. In this case, Hebb's rule can be written as

$$\hat{\mathbf{x}}_{t+1} = \mathbf{x}_t + \eta \hat{\mathbf{Y}}^\top \left(\hat{\mathbf{Y}} \mathbf{x}_t\right); \mathbf{x}_{t+1} = \hat{\mathbf{x}}_{t+1} / \left\|\hat{\mathbf{x}}_{t+1}\right\|_2$$

Notice that the stochastic estimate of the covariance matrix $\hat{\boldsymbol{\Sigma}}$ is never explicitly computed.

# 3. The Parallel Deflation Algorithm

**Algorithm overview.** Our framework distributes the computation of $K$ principal components across $K$ distinct workers, where worker $k$ is responsible for computing the $k$-th principal component. Our innovation lies in reformulating the traditional deflation process for distributed settings. The conventional sequential deflation requires the $k$-th eigenvector computation to wait for the completion of all previous $k-1$ eigenvectors, creating a strict dependency chain that inhibits parallelization. We overcome this limitation through an iterative two-phase approach:

–*Initial Estimation*: Each worker $k$ computes an approximate version of the $k$-th deflated matrix using preliminary estimates of the first $k-1$ eigenvectors from other workers.

–*Iterative Refinement*: Workers computing the first $k-1$ eigenvectors continuously provide updated estimates to worker $k$, enabling progressive refinement of the deflated matrix and the $k$-th eigenvector estimation.

---

**Algorithm 1** Parallel Deflation

---

**Require:** $\boldsymbol{\Sigma} \in \mathbb{R}^{d \times d}$; # of workers $K$; sub-routine for top eigenvector $\mathtt{PCA}(\cdot, \cdot, \cdot)$; # of iterations $T$; global communication rounds $L \geq K$.

**Ensure:** Approximate eigenvectors $\{\mathbf{v}_k\}_{k=1}^K$.

1: **for** $k = 1, \ldots, K$ **do**
2:      Randomly initialize $\hat{\mathbf{v}}_{k,\mathtt{init}}$ with unit norm;
3: **end for**
4: **for** $\ell = 1, \ldots, L$ **do**
5:      **parfor** $k = 1, \ldots, K$ **do**
6:          **if** $k \leq \ell$ **then**
7:              Receive $\mathbf{v}_{1,\ell-1}, \ldots, \mathbf{v}_{k-1,\ell-1}$
8:              $\boldsymbol{\Delta}_{k',\ell} = \mathbf{v}_{k',\ell-1} \mathbf{v}_{k',\ell-1}^\top \boldsymbol{\Sigma} \mathbf{v}_{k',\ell-1} \mathbf{v}_{k',\ell-1}^\top$
9:              $\boldsymbol{\Sigma}_{k,\ell} = \boldsymbol{\Sigma} - \sum_{k'=1}^{k-1} \boldsymbol{\Delta}_{k',\ell}$
10:             $\mathbf{v}_{k,\ell} \leftarrow \mathtt{Top1}\left(\boldsymbol{\Sigma}_{k,\ell}, \mathbf{v}_{k,\ell-1}, T\right)$
11:             Broadcast $\mathbf{v}_{k,\ell}$
12:          **else**
13:              $\mathbf{v}_{k,\ell} := \hat{\mathbf{v}}_{k,\mathtt{init}}$;
14:          **end if**
15:      **end parfor**
16: **end for**
17: **return** $\{\mathbf{v}_{k,L}\}_{k=1}^K$

---

This parallel approach eliminates the need for worker $k$ to wait for the complete convergence of the first $k-1$ eigenvectors before beginning its computation. The complete specification of our parallel deflation algorithm is presented in Algorithm 1. We detail the explanation below:

The computation process is divided into $L$ communication rounds (**Line 4**). In the $\ell$th communication round, the $k$th worker will compute an approximation of the $k$th principal component $\mathbf{v}_{k,\ell}$

by running their own sub-routine in parallel, following the rule that the $k$th worker only deflates its matrix and starts computing the $k$th principal component after the first $k-1$ workers have computed some rough estimation of the first $k-1$ principal components (**Lines 7-10**). Therefore, in the $\ell$th communication round, there can be two scenarios for worker $k$: $i$) if $\ell < k$, this means that not all of the first $k-1$ workers have computed some approximation of their own principal component. Therefore, worker $k$ does not deflate the matrix and output $\mathbf{v}_{k,\ell} = \hat{\mathbf{v}}_{k,\texttt{init}}$; $ii$) If $\ell \geq k$, then the first $k-1$ workers have at least computed one approximation of their own principal component. In this case, worker $k$ deflates the matrix using the most updated vectors $\mathbf{v}_{1,\ell-1}, \ldots \mathbf{v}_{k-1,\ell-1}$ (**Line 7**), compute its approximation of the $k$th principal component by calling the $\texttt{Top1}(\cdot)$ on the deflated matrix starting from its output in the previous communication round (**Line 10**), and then broadcast the current approximation to the other workers for the next communication round (**Line 11**). An illustration of the algorithm is given in Figure 1.

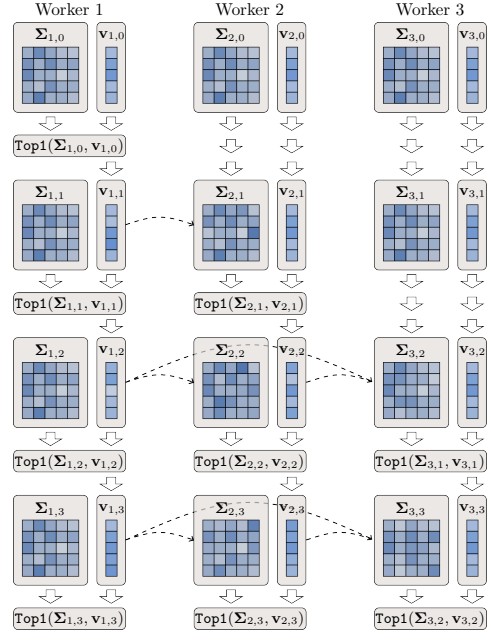

Figure 1: Illustration of the parallel deflation algorithm.

**Extension to Stochastic PCA.** The algorithm described above can be applied to the case where the covariance matrix is either known or can be efficiently estimated. However, in many machine learning scenarios, the covariance matrix may not be direcly accessible. For instance, when data drawn from an underlying distribution comes in a streaming fashion [80], the traditional approach of first estimating the covariance matrix and then solves for its eigenvector is no longer efficient. Moreover, for large datasets that contains hundreds of thousands of features, it is impossible to compute or even store the covariance matrix [33, 34]. In these cases, our algorithm can be adapted to estimate the principal components in a stochastic fashion.

Let $\hat{\mathbf{Y}}$ denote the mini-batch that the algorithm receives in the $t$th iteration. Starting from **Line 8**, whose major computation burden is on $\mathbf{v}_{k',\ell-1}^\top \boldsymbol{\Sigma} \mathbf{v}_{k',\ell-1}$, we notice that the covariance matrix is estimated as $\boldsymbol{\Sigma} \approx \hat{\boldsymbol{\Sigma}} = \hat{\mathbf{Y}}^\top \hat{\mathbf{Y}}$. In this case, we have:

$$\mathbf{v}_{k',\ell-1}^\top \hat{\boldsymbol{\Sigma}} \mathbf{v}_{k',\ell-1} = \mathbf{v}_{k',\ell-1}^\top \hat{\mathbf{Y}}^\top \hat{\mathbf{Y}} \mathbf{v}_{k',\ell-1}$$
$$= \|\hat{\mathbf{Y}} \mathbf{v}_{k',\ell-1}\|_2^2.$$

Therefore, each $\boldsymbol{\Delta}_{k',\ell}$ in Algorithm 1 can be written as $\boldsymbol{\Delta}_{k',\ell} = \|\hat{\mathbf{Y}} \mathbf{v}_{k',\ell-1}\|_2^2 \mathbf{v}_{k',\ell-1} \mathbf{v}_{k',\ell-1}^\top$. Thus, **Line 9** becomes:

$$\boldsymbol{\Sigma}_{k,\ell} \approx \hat{\boldsymbol{\Sigma}}_k = \hat{\mathbf{Y}}^\top \hat{\mathbf{Y}} - \sum_{k'=1}^{k-1} \|\hat{\mathbf{Y}} \mathbf{v}_{k',\ell-1}\|_2^2 \mathbf{v}_{k',\ell-1} \mathbf{v}_{k',\ell-1}^\top.$$

This new form of $\boldsymbol{\Sigma}_{k,\ell}$ allows an efficient estimation of the matrix-vector product $\boldsymbol{\Sigma}_{k,\ell} \mathbf{x}$, as in:

$$\hat{\lambda}_{k'} = \|\hat{\mathbf{Y}} \mathbf{v}_{k'}\|_2^2; \quad \forall k' \in [k-1]; \quad \boldsymbol{\Sigma}_{k,\ell} \mathbf{x} = \hat{\mathbf{Y}}^\top \hat{\mathbf{Y}} \mathbf{x}_t - \sum_{k'=1}^{k-1} \hat{\lambda}_{k'} \left(\mathbf{v}_{k',\ell-1}^\top \mathbf{x}_t\right) \cdot \mathbf{v}_{k',\ell-1}. \tag{4}$$

In the current form of Algorithm 1, the computation of **Lines 8-9** takes $O\left(Kd^2\right)$ time. Moreover, when calling the $\texttt{Top1}$ function in **Lines 10**, any matrix-vector multiplication $\boldsymbol{\Sigma}_{k,\ell} \mathbf{x}$ will take $O\left(d^2\right)$ time. Notice that in (4), the complexity of computing each $\mathbf{Y} \mathbf{v}_{k'}$ is $O\left(nd\right)$. Thus computing $\hat{\lambda}_{k'}$ takes $O\left(nd\right)$. In total,(4) has a complexity of $O\left(Knd\right)$. In (4), computing the first term $\hat{\mathbf{Y}}^\top \hat{\mathbf{Y}} \mathbf{x}$ involves computing first $\mathbf{y}_t = \hat{\mathbf{Y}} \mathbf{x}_t$, which takes $O\left(nd\right)$, and then $\hat{\mathbf{Y}}^\top \mathbf{y}_t$, which also takes $O\left(nd\right)$. Thus, computing the first term $\hat{\mathbf{Y}}^\top \hat{\mathbf{Y}} \mathbf{x}_t$ takes $O\left(nd\right)$ in total. For the second term, each summand takes $O\left(d\right)$ to compute, giving the complexity of computing the second term as $O\left(kd\right)$. Therefore,

each iteration of (4) takes $O((n + k)d)$. This implies a saving in the computation cost, since in this case, $n$ is the batch size and can be much smaller than $d$. The complete algorithm in the stochastic setting is given in Algorithm 2 in the Appendix.

**Communication Analysis:** Let $C_{\text{comm}}$ be the time for one all-reduce operation across workers. The total communication cost per iteration is:

$$T_{\text{comm}} = \frac{1}{2}K(K-1)C_{\text{comm}} \cdot d \tag{5}$$

For high-latency environments, $C_{\text{comm}}$ can be large, resulting in a larger communication cost. In this case, one could use a larger $T$, leading to a smaller communication cost (see Appendix D). Additionally, we should notice that the communication can happen in parallel as the local computation of the eigenvectors. From this perspective, our algorithm also has the potential to be extended to an asynchronous version, where the eigenvectors are updated asynchronously, resulting in a non-blocking computation on each worker.

**Connection with EigenGame.** The EigenGame [33] considers the problem of solving the top-$K$ eigenvectors of a matrix as a game between $K$ players, with the $k$th player solving $\mathbf{v}_k$ by maximizing its utility: $\mathbf{v}_k = \arg\max_{\mathbf{v}:\|\mathbf{v}\|_2=1} \mathcal{U}_k(\mathbf{v} \mid \mathbf{v}_1, \dots \mathbf{v}_{k-1})$, where:

$$\mathcal{U}_k\left(\mathbf{v} \mid \{\mathbf{v}_{k'}\}_{k'=1}^{k-1}\right) = \mathbf{v}^\top \boldsymbol{\Sigma} \mathbf{v} - \sum_{k'=1}^{k-1} \frac{\left(\mathbf{v}_{k'}^\top \boldsymbol{\Sigma} \mathbf{v}\right)^2}{\mathbf{v}_{k'}^\top \boldsymbol{\Sigma} \mathbf{v}_{k'}}. \tag{6}$$

Similarly, the deflation algorithm in (2) also bears a game formulation, where the utility of the $k$th player is given by:

$$\mathcal{V}_k\left(\mathbf{v} \mid \{\mathbf{v}_{k'}\}_{k'=1}^{k-1}\right) = \mathbf{v}^\top \left(\boldsymbol{\Sigma} - \sum_{k'=1}^{k-1} \mathbf{v}_{k'}\mathbf{v}_{k'}^\top \boldsymbol{\Sigma}\mathbf{v}_{k'}\mathbf{v}_{k'}^\top\right)\mathbf{v} = \mathbf{v}^\top \boldsymbol{\Sigma} \mathbf{v} - \sum_{k'=1}^{k-1} \mathbf{v}_{k'}^\top \boldsymbol{\Sigma}\mathbf{v}_{k'} \cdot \left(\mathbf{v}_{k'}^\top \mathbf{v}\right)^2. \tag{7}$$

It should be noted that both the EigenGame utility $\mathcal{U}_k$ and the deflation utility $\mathcal{V}_k$ depend on only the policy of the first $k-1$ players. Moreover, when the first $k-1$ players recovers the top-$(k-1)$ eigenvectors exactly, we shall have:

$$\mathcal{V}_k\left(\mathbf{v} \mid \{\mathbf{u}_{k'}^\star\}_{k'=1}^{k-1}\right) = \mathbf{v}^\top \boldsymbol{\Sigma} \mathbf{v} - \sum_{k'=1}^{k-1} \lambda_{k'}^\star \left(\mathbf{v}^\top \mathbf{u}_{k'}^\star\right)^2 = \mathcal{U}_k\left(\mathbf{v} \mid \{\mathbf{u}_{k'}^\star\}_{k'=1}^{k-1}\right).$$

To this end, we can also show that the set of true eigenvectors $\{\mathbf{u}_k^\star\}_{k=1}^K$ is the unique strict Nash Equilibrium defined by the utilities in (7). The proof of Theorem 1 is deferred to Appendix B.

**Theorem 1.** *Assume that the covariance matrix $\boldsymbol{\Sigma}$ has positive and strictly decreasing eigenvalues $\lambda_1^\star > \cdots > \lambda_K^\star > 0$. Then, $\{\mathbf{u}_k^\star\}_{k=1}^K$ is the unique strict Nash Equilibrium defined by the utilities in (7) up to sign perturbation, i.e., replacing $\mathbf{u}_k^\star$ with $-\mathbf{u}_k^\star$.*

# 4. Convergence Guarantee for the Parallel Deflation Algorithm

We provide a convergence guarantee for the parallel deflation algorithm in Algorithm 1. The pivot of the convergence analysis will be to track the dynamics of $\{\boldsymbol{\Sigma}_{k,\ell}\}_{k=1}^K$ and $\{\mathbf{v}_{k,\ell}\}_{k=1}^K$ as $\ell$ increases. The dynamics of the two sequences from Algorithm 1 can be compactly represented as:

$$\boldsymbol{\Sigma}_{k,\ell} = \boldsymbol{\Sigma} - \sum_{k'=1}^{k-1} \mathbf{v}_{k',\ell-1}\mathbf{v}_{k',\ell-1}^\top \boldsymbol{\Sigma}\mathbf{v}_{k',\ell-1}\mathbf{v}_{k',\ell-1}^\top; \quad \mathbf{v}_{k,\ell} = \texttt{Top1}\left(\boldsymbol{\Sigma}_{k,\ell}, \mathbf{v}_{k,\ell-1}\right); \quad \forall \ell \geq k.$$

Here, we embed the number of solver steps $T$ in the property of the abstract local solver $\texttt{Top1}(\cdot)$. Indeed, if $\texttt{Top1}(\cdot)$ returns the exact top eigenvector of the input matrix every time it is called, then we can easily see that $\mathbf{v}_{k,\ell} = \mathbf{u}_k^\star$ for all $\ell \geq k$. When $\texttt{Top1}(\cdot)$ returns an inexact estimate of the input matrix sequentially, i.e., worker $k$ waits until the top-$(k-1)$ worker no longer improves the estimation of the top-$(k-1)$ eigenvectors, the error is analyzed by [84].

Our scenario is further complicated by the continuous improvement of the eigenvector estimates used to deflate the matrix: in each communication round, the $\texttt{Top1}(\cdot)$ function, called by worker $k$, will start at the estimate of the top eigenvector of the deflated matrix in the previous round but will be fitted to the top eigenvector of the deflated matrix in the current round. Our convergence analysis tackles this complicated dynamic by utilizing the following abstraction of the $\texttt{Top1}(\cdot)$ sub-routine.

**Assumption 1.** *Let $\hat{\mathbf{\Sigma}} \in \mathbb{R}^{d \times d}$ be a real symmetric matrix. Let $\lambda^\star$ be its eigenvalue with the largest absolute value, and let $\mathbf{u}^\star$ be the corresponding eigenvector of $\lambda^\star$. We assume that there exists a real value $\mathcal{F}\left(\hat{\mathbf{\Sigma}}\right) \in (0,1)$ that depends on $\hat{\mathbf{\Sigma}}$ such that for any $\mathbf{x}_0 \in \mathbb{R}^d$, $\mathrm{Top1}(\cdot)$ satisfies:*

$$\left\| \mathrm{Top1}\left(\hat{\mathbf{\Sigma}}, \mathbf{x}_0\right) - \mathbf{u}^\star \right\|_2 \leq \mathcal{F}\left(\hat{\mathbf{\Sigma}}\right) \left\| \mathbf{x}_0 - \mathbf{u}^\star \right\|_2 .$$

Assumption 1 can be easily guaranteed. as long as the $\mathrm{Top1}(\cdot)$ algorithm enjoys a non-asymptotic convergence to the top eigenvector; see the Related Works section above. With Assumption 1, the convergence of Algorithm 1 is given by the following theorem.

**Theorem 2.** *Assume that Assumption 1 holds, and let $\mathcal{F}_k = \max_{\ell \geq k} \mathcal{F}\left(\hat{\Sigma}_{k,\ell}\right)$. Let $W_{-1}\left(\cdot\right)$ be the Lambert-W function in the $-1$ branch[1], and define for $a > 0$:*

$$\hat{W}\left(a\right) = \begin{cases} -W_{-1}\left(-a\right) & \text{if } a \in (0, e^{-1}) \\ 1 & \text{if } a \in [e^{-1}, \infty) \end{cases}$$

*Let $\{m_k\}_{k=0}^K$ be a sequence of numbers denoting the convergence rates of recovering the $K$ eigenvectors, where $m_k = \max\left\{\mathcal{F}_k, \frac{1}{k} + \frac{k-1}{k}m_{k-1}\right\}$ and $m_0 = \mathcal{F}_1$ as a dummy starting point. Let $\{s_k\}_{k=1}^n$ be a sequence of integers denoting the starting communication round where the $K$ eigenvectors' error recovery enters the linear convergence phase, respectively. To be more specific, let $s_1 = 1$ and for all $k \in [K-1]$ and $k' \in [k]$:*

$$s_{k+1} \geq \max\left\{ \frac{\hat{W}\left(m_k \log {}^1\!/m_k\right)}{\log {}^1\!/m_k}, \frac{km_k + 1}{1 - m_k} \right\} + \frac{\hat{W}\left(\frac{\lambda^\star_{k+1} - \lambda^\star_{k+2}}{12k\lambda^\star_{k'}}\left(\log {}^1\!/m_k\right)^2\right)}{\log {}^1\!/m_{k'}} + s_{k'}. \tag{8}$$

*Then, we have that the following holds for all $k \in [K]$*

$$\left\| \mathbf{v}_{k,\ell} - \mathbf{u}^\star_k \right\|_2 \leq 6\left(\ell - s_k + 2\right) m_k^{\ell - s_k + 1}; \quad \forall \ell \geq s_k - 1. \tag{9}$$

In words, Theorem 2 says that starting from the $s_k$th communication round, the recovery error of the $k$th eigenvector converges according to a nearly-linear convergence rate given in (9). However, the convergence starting point $s_k$ for the $k$th eigenvector must be later than the convergence starting point for the $1, \ldots, k-1$th eigenvector for a number of communication rounds. This delay in the convergence starting point is characterized in (8). Intuitively, the starting point $s_k$ denotes the index of the communication round where the top-$(k-1)$ eigenvectors have been estimated accurately enough for the $k$th worker to make positive progress.

**Remark 1.** By the definition that $m_k = \max\left\{\mathcal{F}_k, \frac{1}{k} + \frac{k-1}{k}m_{k-1}\right\}$, one could see that $m_k < 1$ since $\mathcal{F}_k < 1$ for all $k \in [K]$. The convergence rate in (9) involves the product of a linear term $\ell - s_k + 2$ and an exponential term $m_k^{\ell - s_k + 1}$. When $\ell$ is large enough, $m_k^{\ell - s_k + 1}$ decays at a much faster speed than the increase of $\ell - s_k + 2$, thus giving a nearly-linear convergence rate.

**Remark 2.** Upper bound on the separation between the $s_k$'s. By using the inequality that $W_{-1}(e^{-u-1}) \geq -1 - \sqrt{2u} - u$ [85], we could obtain that $\hat{W}(a) \leq \log {}^1\!/a + \sqrt{2(\log {}^1\!/a - 1)} + 1$ when $a \in (0, e^{-1})$. Therefore, we can conclude that $\hat{W} = O(\max\{1, \log {}^1\!/a\})$. Notice that (8) requires that $s_{k+1}$, the starting point of the linear convergence for the error of $\mathbf{v}_{k+1,\ell}$, must be later than $s_1, \ldots, s_k$ for some steps. Using the bound of $\hat{W}(a)$, one could simplify (8) to:

$$s_{k+1} \geq s_k + O\left(\max\left\{\left(\log \frac{1}{m_k}\right)^{-1}\left(1 + \log \frac{k\lambda^\star_k}{\lambda^\star_{k+1} - \lambda^\star_{k+2}}\right), \frac{km_k + 1}{1 - m_k}\right\}\right).$$

This simplification implies that a smaller $m_k$ would cause a smaller decay between $s_k$ and $s_{k+1}$. Since $m_k$ depends on $\mathcal{F}_k$, a smaller $m_k$ can be achieved by doing more local steps in the call to $\mathrm{Top1}(\cdot)$. However, since the decay between $s_k$ and $s_{k+1}$ is measured in terms of the communication rounds, doing more local steps also increases the computation time per round in the delayed periods.

**Sketch of Proof.** The key challenge in proving Theorem 2 lies in handling the dynamic, asynchronous nature of our algorithm. Unlike sequential deflation, where each principal component

---

[1]The Lambert-W function in the $-1$-branch is defined as the inverse of the function $f(x) = xe^x$ when $x \in (-\infty, -1)$.

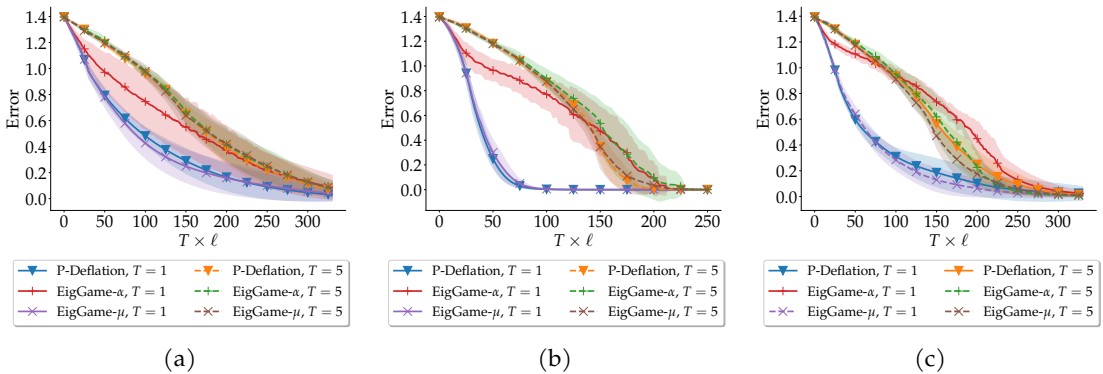

Figure 2: Comparison of the convergence behavior of parallel deflation, EigenGame-$\alpha$, and EigenGame-$\mu$ in deterministic setting on (a). synthetic dataset with power-law decaying eigenvalues, (b). synthetic dataset with exponentially decaying eigenvalues, and (c). MNIST dataset.

is computed after the previous ones have converged and stays fixed, our method deals with simultaneous updates of all principal components. This requires careful analysis of how errors propagate and accumulate across different workers. To start, we derive upper bounds of the per-iteration difference between the actual deflated matrix $\boldsymbol{\Sigma}_k$ and the ideal deflated matrix $\boldsymbol{\Sigma}_k^*$ and the difference between the estimated eigenvector $\mathbf{v}_{k,\ell}$ and the ground-truth eigenvector $\mathbf{u}_k^*$. Notice that these two upper bounds are inter-dependent. We apply Davis-Kahan Sin$\Theta$ Theorem [86] to derive the bound between the matrices' top-eigenvectors based on the matrix differences. Next, we carefully choose a convergence starting point $s_k$ for each eigenvector. Construct two simpler two-dimensional sequences $\{B_{k,\ell}\}$ and $\{G_{k,\ell}\}$ starting from $s_k$'s that upper bound these differences. Lastly, we unroll the bounds on $\{B_{k,\ell}\}$ and $\{G_{k,\ell}\}$ to arrive at a closed form upper bound on error of the estimated eigenvector $\mathbf{v}_{k,\ell}$. The detailed proof is deferred to Appendix C.

## 5. Experiments

In this section, we experimentally verify the performance of the parallel deflation algorithm.

**Baseline algorithms.** We compared the parallel deflation algorithm with power iteration as the Top1 subroutine with the distributed version of EigenGame-$\alpha$ [33] and EigenGame-$\mu$ [34]. EigenGame-$\alpha$ was proposed as a sequential principal component recovery algorithm and can be adapted as a distributed algorithm. Both EigenGame-$\alpha$ and EigenGame-$\mu$ are restricted to the case of one iteration of update per communication round. We modified their algorithm to generalize to multiple iterations of update in Algorithm 3 and Algorithm 4 in Appendix F. As in the implementation of [33] and [34], we do not project the utility gradient to the unit sphere.

**Evaluation Metric.** We evaluate the performance of the three algorithms by computing how close the recovered principal component is to the true eigenvector of the covariance matrix. For the set of true principal components $\{\mathbf{u}_k^\star\}_{k=1}^K$ and a set of recovered principal component $\{\mathbf{v}_k\}_{k=1}^K$, we use the following metric to compute the approximation error[2]

$$\mathcal{E}\left(\{\mathbf{u}_k^\star\}_{k=1}^K, \{\mathbf{v}_k\}_{k=1}^K\right) = \left(\frac{1}{K}\sum_{k=1}^K \min_{s\in\{\pm1\}} \|\mathbf{u}_k^\star - s\cdot\mathbf{v}_k\|_2^2\right)^{\frac{1}{2}} \tag{10}$$

**Deterministic Experiments.** For synthetic experiments, we choose the number of features $d = 1000$, which gives the covariance matrices $\boldsymbol{\Sigma} \in \mathbb{R}^{1000\times1000}$. We consider $\boldsymbol{\Sigma}$ generated with two different eigenvalue spectra: $i$) a power-law decaying spectrum $\lambda_k^\star = \frac{1}{\sqrt{k}}$, and $ii$) an exponential decaying spectrum $\lambda_k^\star = \frac{1}{1.1^k}$. We choose the number of local updates in each communication round $T$ to be $T \in \{1, 5\}$. We ran parallel deflation, EigenGame-$\alpha$, and EigenGame-$\mu$ to recover the top-30 eigenvectors ($K = 30$). For each setting, we run 10 trials with different random initialization.

---

[2]We choose this metric instead of the Longest Correct Eigenvector Streak [33] to study the precise dynamic of the recovery error as the number of computations steps increases.

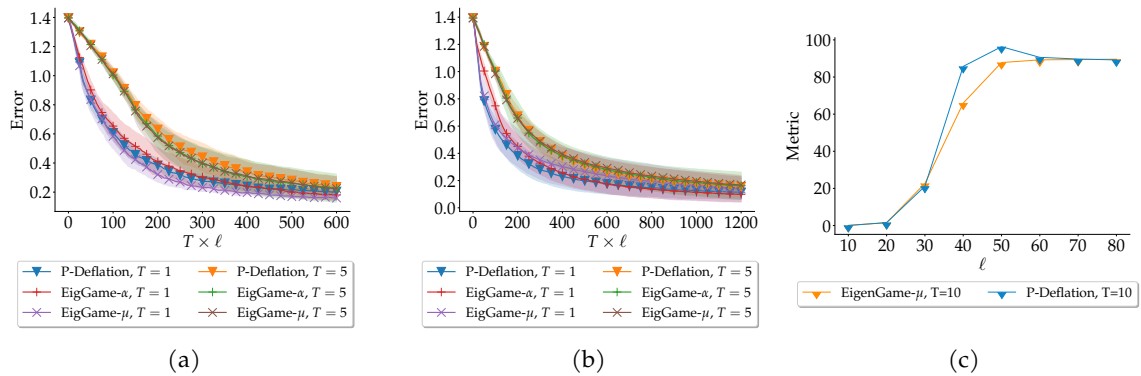

Figure 3: Comparison of the convergence behavior of parallel deflation, EigenGame-$\alpha$, and EigenGame-$\mu$ in stochastic setting on (a). synthetic dataset with power-law decaying eigenvalues, (b). MNIST dataset, and (c) ImageNet dataset.

Figure 2a presents the convergence behavior of the three algorithms with $T \in \{1, 5\}$ on the synthetic matrix with $\lambda_k^\star = \frac{1}{\sqrt{k}}$. Both EigenGame-$\mu$ and parallel deflation demonstrate stable convergence to a low error value under the case of $T = 1$ and $T = 5$, with parallel deflation converging slightly slower than EigenGame-$\mu$ in the first 200 total steps, and then arriving at a lower error than EigenGame-$\mu$ in the last 100 total steps. Figure 2b presents the result on the synthetic matrix with $\lambda_k^\star = \frac{1}{1.1^k}$, with similar conclusions. In both Figure 2a and Figure 2b, the setting of $T = 1$ shows a faster convergence than $T = 5$. This is because $T = 1$ allows more communication in a fixed number of total steps $T \times L$, which keeps the deflated matrices of each local worker to be better updated.

We also use the real-world dataset of MNIST, $\mathbf{Y} \in \mathbb{R}^{60000 \times 784}$ in Figure 2c. We choose $T \in \{1, 5\}$ and aim at recovering the top-30 eigenvectors. We observe a similar convergence behavior of the three algorithms as above. For EigenGame-$\alpha$, the case $T = 1$ converges even slower than the case $T = 5$. We hypothesize that this is because one local iteration is not sufficient for the top eigenvector solvers to provide an accurate enough estimate for the following solvers to make positive progress.

**Stochastic Setting.** We generate $\mathbf{\Sigma}$ with power-law decaying spectrum as in the deterministic experiments. We sample I.I.D. samples from $\mathcal{N}(\mathbf{0}, \mathbf{\Sigma})$ and pass the sampled data batches to parallel deflation and EigenGame in a streaming fashion. We use a decaying step size for all three algorithms, and the result is given in Figure 3a. In this setting, parallel deflation shows a slightly worse performance than the two EigenGame algorithms. We hypothesize that this is because parallel deflation is more sensitive to the step size tuning in the stochastic case. In Figure 3b, we plot the performance of parallel deflation and EigenGame in the stochastic setting of the MNIST dataset. Parallel deflation achieves similar performance to EigenGame-$\mu$, with a slightly faster convergence speed in the early phase of the algorithm.

We also test the performance of parallel deflation on the ImageNet dataset [36] that contains $n = 1.2M$ images of $d = 50176$ pixels. Due to the size of $\mathbf{Y}$, it was not possible in our set up to compute or store the covariance matrix on a single device. We use both parallel deflation and EigenGame-$\mu$ to compute the top-10 eigenvector of the dataset. Since no "ground-truth" principal component is known, we can use an aggregation of the terms $\mathbf{v}_k^\top \mathbf{\Sigma} \mathbf{v}_k$ as a metric to evaluate the quality of the solved principal components. To follow the internal hierarchy of the eigenvectors that the leading eigenvectors are free to explore more space and thus are expected to attain a larger $\mathbf{v}^\top \mathbf{\Sigma} \mathbf{v}$, we penalize terms with larger index with a discounting factor. This result in the following metric:

$$\mathcal{M}\left(\{\mathbf{v}_k\}_{k=1}^K\right) = \sum_{k=1}^K \frac{1}{k} \mathbf{v}_k^\top \mathbf{\Sigma} \mathbf{v}_k = \frac{1}{n} \sum_{i=1}^n \sum_{k=1}^K \frac{\left(\bar{\mathbf{y}}_i^\top \mathbf{v}_k\right)^2}{k} \tag{11}$$

Notice that a larger value of $\mathcal{M}(\cdot)$ implies a higher quality of the recovered eigenvectors. Figure 3c shows that in this large scale dataset, our algorithm can keep up with the performance of the state-of-the-art algorithm EigenGame-$\mu$.

## 6. Conclusion

We present an algorithmic framework for computing the principal components in a distributed fashion. We introduce additional parallelism by early-starting the computation of the following eigenvectors based on the initial rough estimation of leading principal components and continuously refining the local deflated matrix based on updated estimated principal components. Our framework has a similar game-theoretic formulation as the EigenGame, while enjoying a nice convergence guarantee even in the distributed case. Future work can focus on empirically examining the potential of using other Top1 subroutines in our parallel deflation algorithm, such as Oja's rule.

## 7. Acknowledgements

This work is supported by NSF CMMI no. 2037545, NSF CAREER award no. 2145629, Welch Foundation Grant no. A22-0307, a Microsoft Research Award, an Amazon Research Award, and a Ken Kennedy Research Cluster award (Rice K2I). AK was supported in part through seed funding from the Ken Kennedy Institute at Rice University.

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

# A. Appendix

# B. Missing Proof from Section 3

*Proof of Theorem 1.* Let $\mathbf{u}_1^\star, \ldots, \mathbf{u}_d^\star$ be the set of eigenvectors of $\mathbf{\Sigma}$, and $\lambda_1^\star, \ldots, \lambda_d^\star$ be the corresponding eigenvalues, potentially with some $\lambda_k = 0$. Recall that in the game formulation of the deflation algorithm, the utility function of the $k$th player is given by

$$\mathcal{V}_k \left( \mathbf{v} \mid \{\mathbf{v}_{k'}\}_{k'=1}^{k-1} \right) = \mathbf{v}^\top \mathbf{\Sigma} \mathbf{v} - \sum_{k'=1}^{k-1} \mathbf{v}_{k'}^\top \mathbf{\Sigma} \mathbf{v}_{k'} \cdot \left( \mathbf{v}_{k'}^\top \mathbf{v} \right)^2$$

and when $\mathbf{v}_{k'} = \mathbf{u}_{k'}^\star$ for $k' \in [k-1]$, we have

$$\mathcal{V}_k\left(\mathbf{v} \mid \{\mathbf{u}_{k'}^\star\}_{k'=1}^{k-1}\right) = \mathbf{v}^\top \boldsymbol{\Sigma} \mathbf{v} - \sum_{k'=1}^{k-1} \lambda_{k'}^\star \cdot \left(\mathbf{v}_{k'}^\top \mathbf{v}\right)^2$$

It should be noted that $\boldsymbol{\Sigma}$ has the eigendecomposition $\boldsymbol{\Sigma} = \sum_{k'=1}^d \lambda_{k'}^\star \mathbf{u}_{k'}^\star \mathbf{u}_{k'}^{\star\top}$. Therefore we can rewrite $\mathbf{v}^\top \boldsymbol{\Sigma} \mathbf{v}$ as $\sum_{k'=1}^d \lambda_{k'}^\star \left(\mathbf{v}^\top \mathbf{u}_{k'}^\star\right)^2$. Thus, the utility becomes

$$\mathcal{V}_k\left(\mathbf{v} \mid \{\mathbf{u}_{k'}^\star\}_{k'=1}^d\right) = \sum_{k'=k}^{k-1} \lambda_{k'}^\star \left(\mathbf{v}^\top \mathbf{u}_{k'}^\star\right)^2$$

Since $\mathbf{u}_1^\star, \ldots, \mathbf{u}_d^\star$ spans $\mathbb{R}^d$ and are mutually orthogonal, we can write $\mathbf{v} = \sum_{j=1}^d \beta_j \mathbf{u}_j^\star$. where $\sum_{j=1}^d \beta_j^2 = 1$. Then we have

$$\mathcal{V}_k\left(\mathbf{v} \mid \{\mathbf{u}_{k'}^\star\}_{k'=1}^d\right) = \sum_{k'=k}^{k-1} \lambda_{k'}^\star \left(\sum_{j=1}^d \beta_j \mathbf{u}_j^{\star\top} \mathbf{u}_{k'}^\star\right)^2 = \sum_{k'=k}^{k-1} \lambda_{k'}^\star \beta_{k'}^2$$

Since $\lambda_k$'s are strictly decreasing and positive, we must have that the maximum of $\mathcal{V}_k\left(\mathbf{v} \mid \{\mathbf{u}_{k'}^\star\}_{k'=1}^d\right)$ is only attained when $\beta_k^2 = 1$, which implies that $\mathbf{v} = \pm\mathbf{u}_k^\star$ will be the only optimal policies for player $k$.

The uniqueness can be shown by induction. To start, we notice that $\mathcal{V}_1$ does not depend on the policy of the other plays. Therefore, the only optimal policy for player 1 is $\mathbf{v}_1 = \pm\mathbf{u}_1^\star$. This shows the base case. Now, assume that within the top-$(k-1)$ players, the optimal policies are $\mathbf{v}_{k'} = \pm\mathbf{u}_{k'}^\star$ for $k' \in [k-1]$. By the formulation of the utility functions, these optimal policies are not affected by the policy of player $k, \ldots, K$. Moreover, it should be noted that the utility of player $k$ only depends on the top-$(k-1)$ players. Therefore, the optimal policy for player $k$ must be $\mathbf{v}_k = \pm\mathbf{u}_k^\star$. This finishes the inductive step and completes the proof. □

## C. Missing Proof from Section 4

We first introduce a tool that we will utilize in the proof in this section.

**Lemma 1** ($\sin\Theta$ Theorem [86]). *Let $\mathbf{M}^* \in \mathbb{R}^{d\times d}$ and let $\mathbf{M} = \mathbf{M}^* + \mathbf{H}$. Let $\mathbf{a}_1^*$ and $\mathbf{a}_1$ be the top eigenvectors of $\mathbf{M}^*$ and $\mathbf{M}$, respectively. Then we have:*

$$\sin\angle\{\mathbf{a}_1^*, \mathbf{a}_1\} \leq \frac{\|\mathbf{H}\|_2}{\min_{j\neq k} |\sigma_k^* - \sigma_j|}.$$

### C.1. Proof of Theorem 2

Define $\boldsymbol{\Sigma}_k^\star = \sum_{k=1}^d \lambda_k^\star \mathbf{u}_k^\star \mathbf{u}_k^{\top\star}$ as the "ground-truth" deflation matrix. Recall that the parallel deflation algorithm executes

$$\boldsymbol{\Sigma}_{k,\ell} = \boldsymbol{\Sigma} - \sum_{k'=1}^{k-1} \mathbf{v}_{k',\ell-1}\mathbf{v}_{k',\ell-1}^\top \boldsymbol{\Sigma} \mathbf{v}_{k',\ell-1}\mathbf{v}_{k',\ell-1}^\top; \quad \mathbf{v}_{k,\ell} = \texttt{Top1}\left(\boldsymbol{\Sigma}_{k,\ell}, \mathbf{v}_{k,\ell-1}\right) \tag{12}$$

Let $\mathbf{u}_{k,\ell}$ denote the top eigenvector of $\boldsymbol{\Sigma}_{k,\ell}$. In particular, it suffices to show that the quantity $\|\mathbf{v}_{k,\ell} - \mathbf{u}_k^\star\|_2^2$ decreases as $\ell$ increases. Combining Assumption 1 and the definition that $\mathcal{F}_k = \max_{\ell\geq k} \mathcal{F}(\boldsymbol{\Sigma}_{k,\ell})$, we have that

$$\|\mathbf{v}_{k,\ell} - \mathbf{u}_{k,\ell}\|_2 \leq \mathcal{F}_k \|\mathbf{v}_{k,\ell-1} - \mathbf{u}_{k,\ell}\|_2 \tag{13}$$

We could upper bound $\|\mathbf{v}_{k,\ell-1} - \mathbf{u}_{k,\ell}\|_2$ using

$$\begin{aligned}
\|\mathbf{v}_{k,\ell-1} - \mathbf{u}_{k,\ell}\|_2 &\leq \|\mathbf{v}_{k,\ell-1} - \mathbf{u}_{k,\ell-1}\|_2 + \|\mathbf{u}_{k,\ell} - \mathbf{u}_{k,\ell-1}\|_2 \\
&\leq \|\mathbf{v}_{k,\ell-1} - \mathbf{u}_{k,\ell-1}\|_2 + \|\mathbf{u}_{k,\ell} - \mathbf{u}_k^\star\|_2 + \|\mathbf{u}_{k,\ell-1} - \mathbf{u}_k^\star\|_2
\end{aligned}$$

Combining this upper bound with (13) gives

$$\|\mathbf{v}_{k,\ell} - \mathbf{u}_{k,\ell}\|_2 \leq \mathcal{F}_k \left( \|\mathbf{v}_{k,\ell-1} - \mathbf{u}_{k,\ell-1}\|_2 + \|\mathbf{u}_{k,\ell} - \mathbf{u}_k^\star\|_2 + \|\mathbf{u}_{k,\ell-1} - \mathbf{u}_k^\star\|_2 \right) \tag{14}$$

Moreover, the triangle inequality implies that

$$\|\mathbf{v}_{k,\ell} - \mathbf{u}_k^\star\|_2 \leq \|\mathbf{v}_{k,\ell} - \mathbf{u}_{k,\ell}\|_2 + \|\mathbf{u}_{k,\ell} - \mathbf{u}_k^\star\|_2 \tag{15}$$

Now, (14) and (15) give a pretty good characterization of the propagation of the errors. It remains to characterize $\|\mathbf{u}_{k,\ell} - \mathbf{u}_k^\star\|_2$ for each $\ell$, and then we can dive into solving the recurrence. A naive bound would be that $\|\mathbf{u}_{k,\ell} - \mathbf{u}_k^\star\|_2 \leq 2$, as $\|\mathbf{u}_{k,\ell}\|_2 = \|\mathbf{u}_k^\star\|_2 = 1$. However, notice that $\mathbf{u}_{k,\ell}$ is the top eigenvector of $\mathbf{\Sigma}_{k,\ell}$ and $\mathbf{\Sigma}_k^\star$, respective. Thus, we can invoke the Davis-Kahan Theorem to obtain a tighter bound. This property is given by Lemma 2, whose proof is deferred to Appendix C.2.

**Lemma 2.** *Assume that $1 = \lambda_1^\star > \lambda_2^\star > \dots$. If the following inequality holds for some $c_0 > 1$*

$$\sum_{k'=1}^{k-1} \lambda_{k'}^\star \|\mathbf{u}_{k'}^\star - \mathbf{v}_{k',\ell-1}\|_2 \leq \frac{c_0 - 1}{4c_0} \left( \lambda_k^\star - \lambda_{k+1}^\star \right) \tag{16}$$

*then we have that*

$$\|\mathbf{u}_{k,\ell} - \mathbf{u}_k^\star\|_2 \leq \frac{4c_0}{\lambda_k^\star - \lambda_{k+1}^\star} \sum_{k'=1}^{k-1} \lambda_{k'}^\star \|\mathbf{u}_{k'}^\star - \mathbf{v}_{k',\ell-1}\|_2 \tag{17}$$

Now, we are going to use induction to proceed with the proof. Notice that, in order to control $\|\mathbf{u}_{k,\ell} - \mathbf{u}_k^\star\|_2$ using Lemma 2, one only need to control the recovery error of all previous eigenvectors $\|\mathbf{u}_{k'}^\star - \mathbf{v}_{k',\ell-1}\|_2$, as given in (16). Thus, fix some $k$, we will assume the inductive hypothesis that there exists some $s$ such that for all $\ell \geq s$, we can guarantee (16). For the case of $k = 1$, this is obvious, as the left-hand side of (16) is 0. When $k \geq 1$ and we can gather the conditions as

$$\|\mathbf{v}_{k,\ell} - \mathbf{u}_{k,\ell}\|_2 \leq \mathcal{F}_{k,\ell} \left( \|\mathbf{v}_{k,\ell-1} - \mathbf{u}_{k,\ell-1}\|_2 + \|\mathbf{u}_{k,\ell} - \mathbf{u}_k^\star\|_2 + \|\mathbf{u}_{k,\ell-1} - \mathbf{u}_k^\star\|_2 \right)$$

$$\|\mathbf{u}_k^\star - \mathbf{v}_{k,\ell}\|_2 \leq \|\mathbf{v}_{k,\ell} - \mathbf{u}_{k,\ell}\|_2 + \|\mathbf{u}_{k,\ell} - \mathbf{u}_k^\star\|_2$$

$$\|\mathbf{u}_{k,\ell} - \mathbf{u}_k^\star\|_2 \leq \frac{4c_0}{\lambda_k^\star - \lambda_{k+1}^\star} \sum_{k'=1}^{k-1} \lambda_{k'}^\star \|\mathbf{u}_{k'}^\star - \mathbf{v}_{k',\ell-1}\|_2 ; \quad \forall \ell \geq s_k$$

For simplicity, we let

$$\|\mathbf{v}_{k,\ell} - \mathbf{u}_{k,\ell}\|_2 =: D_{k,\ell}; \quad \|\mathbf{u}_{k,\ell} - \mathbf{u}_k^\star\|_2 =: B_{k,\ell}; \quad \|\mathbf{u}_k^\star - \mathbf{v}_{k,\ell}\|_2 =: G_{k,\ell}$$

Moreover, we let $\mathcal{C}_k = \frac{4c_0}{\lambda_k^\star - \lambda_{k+1}^\star}$. Then the iterates are simplified to

$$D_{k,\ell} \leq \mathcal{F}_k \left( D_{k,\ell-1} + B_{k,\ell} + B_{k,\ell-1} \right)$$

$$G_{k,\ell} \leq D_{k,\ell} + B_{k,\ell}$$

$$B_{k,\ell} \leq \mathcal{C}_k \sum_{k'=1}^{k-1} \lambda_{k'}^\star G_{k',\ell-1}$$

where we set $G_{0,\ell} = 0$ for all $\ell$. Then $G_{k,\ell}$ can be written as

$$G_{k,\ell} \leq \mathcal{F}_k^{\ell-s} D_{k,s} + \sum_{\ell'=s}^{\ell-1} \mathcal{F}_k^{\ell-\ell'} \left( B_{k,\ell'} + B_{k,\ell'-1} \right) + B_{k,\ell}$$

for any $s \in [\ell]$. Here, the first term can be made small as long as we choose a large enough $\ell$. The third term is the unavoidable error propagation. The second term can cause $G_{k,\ell}$ to grow, and needs a careful analysis. To understand the recurrence between $G_{k,\ell}$ and $B_{k,\ell}$, we use the following lemma

**Lemma 3.** *Let $\hat{s}_k$ be given for all $k \in [K]$ such that $1 \leq \hat{s}_1 \leq \dots \leq \hat{s}_K$. Let $s_k \in \mathbb{Z}$ be given for all $k \in [K]$ such that $1 = s_0 \leq s_1 \leq \dots \leq s_K$. Consider the sequence $\{B_{k,\ell}\}_{\ell=\hat{s}_k}^\infty$ and $\{G_{k,\ell}\}_{\ell=s_k-1}^\infty$ for all $k \in [K]$ characterized by the following recurrence*

$$B_{k,\ell} \leq C_k \sum_{k'=1}^{k-1} \lambda_{k'}^\star G_{k',\ell-1}$$

$$\tag{18}$$

$$G_{k,\ell} \leq \mathcal{F}_k^{\ell-s_k+1} D_{k,s_k-1} + \sum_{\ell'=s_k-1}^{\ell-1} \mathcal{F}_k^{\ell-\ell'} \left( B_{k,\ell'} + B_{k,\ell'-1} \right) + B_{k,\ell}$$

Let $m_k = \max\{\mathcal{F}_k, \gamma_{k-1}\}$ for all $k \in [K]$ and $m_0 = -1$. Let $\{\gamma\}_{k=-1}^K$ be given such that $\gamma_{-1} = \gamma_0 = 0$ and $\gamma_k = \frac{1}{k+1} + \frac{k}{k+1} m_k$ for all $k \in [K]$. Define sequences $\{\hat{B}_{k,\ell}\}_{\ell=\hat{s}_k}^\infty$ and $\{\hat{G}_{k,\ell}\}_{\ell=s_k-1}^\infty$ for all $k \in [K]$ as

$$\hat{B}_{k,\ell} = \begin{cases} \min\left\{2, m_{k-1}^{\ell-\hat{s}_k}(\ell - \hat{s}_k + 1)\hat{B}_{k,\hat{s}_k}\right\} & \text{if } \ell > \hat{s}_k \\ C_k \sum_{k'=1}^{k-1} \lambda_{k'}^\star \hat{G}_{k',\hat{s}_k-1} & \text{if } \ell = \hat{s}_k \end{cases}$$

$$\hat{G}_{k,\ell} = \begin{cases} m_k^{\ell-s_k+1}(\ell - s_k + 2)\hat{G}_{k,s_k-1} & \text{if } \ell \geq s_k \\ D_{k,s_k-1} + \hat{B}_{k,s_k-1} + \hat{B}_{k,s_k-2} & \text{if } \ell = s_k - 1 \end{cases} \tag{19}$$

Suppose that $\hat{s}_{k+1} \geq s_k$, and $s_k$ satisfies satisfies $m_{k-1}^{s_k-\hat{s}_k-2} \leq \frac{1}{s_k-\hat{s}_k-1}$ and $s_k \geq \frac{k m_{k-1}}{1-m_{k-1}} + \hat{s}_k + 2$. Moreover, suppose that $\hat{B}_{k,\hat{s}_k} \leq 2$ for all $k \in [K]$. Then the following two conditions hold

1. $\hat{B}_{k,\ell} \geq B_{k,\ell}$ for all $\ell \geq \hat{s}_k$

2. $\hat{G}_{k,\ell} \geq G_{k,\ell}$ for all $\ell \geq s_k - 1$

The proof of Lemma 3 is deferred to Appendix C.3. Lemma 3 implies that under proper condition of $s_k$ and $\hat{s}_k$, we have

$$G_{k,\ell} \leq \hat{G}_{k,\ell}$$
$$\leq \max\{\mathcal{F}_k, \gamma_{k-1}\}^{\ell-s_k+1}(\ell - s_k + 1)\hat{G}_{k,s_k-1}$$
$$= \max\{\mathcal{F}_k, \gamma_{k-1}\}^{\ell-s_k+1}(\ell - s_k + 1)\left(D_{k,s_k-1} + \hat{B}_{k,s_k-1} + \hat{B}_{k,s_k-2}\right)$$

By definition, we have that $D_{k,s_k-1} = \|\mathbf{v}_{k,s_k-1} - \mathbf{u}_{k,s_k-1}\|_2 \leq 2$. Moreover, the definition in (19) gives that $\hat{B}_{k,s_k-1} \leq 2$ and $\hat{B}_{k,s_k-2} \leq 2$. Therefore, we can conclude that

$$G_{k,\ell} \leq 6(\ell - s_k + 1)\max\{\mathcal{F}_k, \gamma_{k-1}\}^{\ell-s_k+1}$$

Now, we go back to the condition of $s_k$ and $\hat{s}_k$. The requirement of $\{s_k\}_{k=0}^K$ and $\{\hat{s}_k\}_{k=1}^K$ can be gathered below

1. $1 = s_0 \leq s_1 \leq \ldots s_K$ and $1 \leq \hat{s}_1 \leq \cdots \leq \hat{s}_K$
2. $\hat{s}_{k+1} \geq s_k$ and $s_k \geq \frac{k m_{k-1}}{1-m_{k-1}} + \hat{s}_k + 2$
3. $m_{k-1}^{s_k} - \hat{s}_k - 2 \leq \frac{1}{s_k-\hat{s}_k-1}$
4. $m_{k-1}^{\ell-\hat{s}_k}(\ell - \hat{s}_k + 1)\sum_{k'=1}^{k-1} \lambda_{k'}^\star \hat{G}_{k',\hat{s}_k-1} \leq \frac{c_0-1}{4c_0}\left(\lambda_k^\star - \lambda_{k+1}^\star\right)$ for all $\ell \geq \hat{s}_k$

where the first three conditions are directly required by Lemma 3, and the fourth condition is required because the upper bound on $B_{k,\ell}$ in (18) hold only when

$$\sum_{k'=1}^{k-1} \lambda_{k'}^\star \|\mathbf{u}_{k'}^\star - \mathbf{v}_{k',\ell-1}\|_2 \leq \frac{c_0-1}{4c_0}\left(\lambda_k^\star - \lambda_{k+1}^\star\right)$$

from Lemma 2. Notice that since $\hat{B}_{k,\hat{s}_k} = C_k \sum_{k'=1}^{k-1} \lambda_{k'}^\star \hat{G}_{k',\hat{s}_k-1}$, enforcing the fourth condition directly implies that $\hat{B}_{k,\hat{s}_k} \leq 2$. Now, we are going to simplify these conditions. A useful tool will be the following lemma, whose proof is provided in Appendix C.4.

**Lemma 4.** Let $m \in (0, 1)$ and $\epsilon \in \mathbb{R}$ be given. Let $g(x) = m^x(x + 1)$, and let $W_{-1}(\cdot)$ be the Lambert-W function. Then

1. When $\epsilon \geq -\frac{1}{em \log m}$, then any $x \geq 0$ satisfies $g(x) \leq \epsilon$

2. When $\epsilon \leq -\frac{1}{em \log m}$, then any $x \geq \frac{1}{\log m} W_{-1}(\epsilon m \log m) - 1$ satisfies $g(x) \leq \epsilon$

Notice that #4 in the conditions above implies that $\|\boldsymbol{\Sigma}_{k,\ell} - \boldsymbol{\Sigma}_k^\star\|_F \leq \frac{c_0-1}{c_0}\left(\lambda_k^\star - \lambda_{k+1}^\star\right)$ and

$$B_{k,\hat{s}_k} = \sum_{k'=1}^{k-1} \lambda_{k'}^\star \hat{G}_{k',\hat{s}_k-1} \leq c_0 - 1$$

Choose $c_0 = 3$ then guarantees that $B_{k,\hat{s}_k} \leq 2$. Now, we aim at simplifying Condition #4 above. To start, we notice that the term $m_{k-1}^{\ell-\hat{s}_k}(\ell - \hat{s}_k + 1)$ achieves global maximum at $\ell - \hat{s}_k = \frac{1}{\log 1/m_{k-1}} - 1$ with value $\frac{1}{\log 1/m_{k-1}} m_{k-1}^{\frac{1}{\log 1/m_{k-1}} - 1}$. Therefore, it suffices to guarantee that

$$\sum_{k'=1}^{k-1} \lambda_{k'}^{\star} \hat{G}_{k',\hat{s}_k-1} \leq -\log m_{k-1} \cdot m_{k-1}^{\frac{1}{\log m_{k-1}} - 1} \cdot \frac{1}{6}\left(\lambda_k^{\star} - \lambda_{k+1}^{\star}\right)$$

From Lemma 3, we have that for $\ell \geq s_k - 1$, $G_{k,\ell} \leq \hat{G}_{k,\ell}$, and

$$\hat{G}_{k,\ell} = m_k^{\ell-s_k+1}(\ell - s_k + 2)\hat{G}_{k,s_k-1}$$

with $\hat{G}_{k,s_k-1} \leq 6$. Therefore, it suffices to guarantee that

$$\sum_{k'=1}^{k-1} \lambda_{k'}^{\star} m_{k'}^{\hat{s}_k-s_{k'}+1}(\hat{s}_k - s_{k'} + 2)\hat{G}_{k,s_{k'}-1} \leq -\log m_{k-1} \cdot m_{k-1}^{\frac{1}{\log m_{k-1}} - 1} \cdot \frac{1}{6}\left(\lambda_k^{\star} - \lambda_{k+1}^{\star}\right)$$

which would be satisfied if we have

$$\lambda_{k'}^{\star} m_{k'}^{\hat{s}_k-s_{k'}+1}(\hat{s}_k - s_{k'} + 2)\hat{G}_{k,s_{k'}-1} \leq -\frac{\lambda_k^{\star} - \lambda_{k+1}^{\star}}{6(k-1)}\log m_{k-1} \cdot m_{k-1}^{\frac{1}{\log m_{k-1}} - 1}$$

Thus, $\hat{s}_k$ must satisfy for all $s_{k'}$

$$m_{k'}^{\hat{s}_k-s_{k'}+1}(\hat{s}_k - s_{k'} + 2) \leq -\frac{\lambda_k^{\star} - \lambda_{k+1}^{\star}}{36\lambda_{k'}^{\star}(k-1)}\log m_{k-1} \cdot m_{k-1}^{\frac{1}{\log m_{k-1}} - 1} \tag{20}$$

With the help of Lemma 4, Condition #3 transfers to

$$s_k \geq \frac{1}{\log m_{k-1}}W_{-1}\left(m_{k-1}\log m_{k-1}\right) + \hat{s}_k + 1$$

Similarly, Condition #4 transfers to

$$\hat{s}_k \geq \frac{1}{\log m_{k'}}W_{-1}\left(-\frac{\lambda_k^{\star} - \lambda_{k+1}^{\star}}{36\lambda_{k'}^{\star}(k-1)}\log m_{k-1} \cdot m_{k-1}^{\frac{1}{\log m_{k-1}} - 1}m_{k'}\log m_{k'}\right) + s_{k'} - 2$$

which can be guaranteed as long as

$$\hat{s}_k \geq \frac{1}{\log m_{k'}}W_{-1}\left(-\frac{\lambda_k^{\star} - \lambda_{k+1}^{\star}}{36k\lambda_{k'}^{\star}}\left(\log m_{k-1}\right)^2 \cdot m_{k-1}^{\frac{1}{\log m_{k-1}}}\right) + s_{k'} - 2$$

Gathering all requirements, we have

$$s_k \geq \max\left\{\frac{1}{\log m_{k-1}}W_{-1}\left(m_{k-1}\log m_{k-1}\right), \frac{(k-1)m_{k-1}+1}{1 - m_{k-1}}\right\} + \hat{s}_k + 1$$

$$\hat{s}_k \geq \frac{1}{\log m_{k'}}W_{-1}\left(-\frac{\lambda_k^{\star} - \lambda_{k+1}^{\star}}{36k\lambda_{k'}^{\star}}\left(\log m_{k-1}\right)^2 \cdot m_{k-1}^{\frac{1}{\log m_{k-1}}}\right) + s_{k'} - 2$$

Plugging $\hat{s}_k$ into the lower bound of $s_k$ shows that the condition in (8) suffice to guarantee that Lemma 3 holds. Thus, we can conclude that

$$\|\mathbf{v}_{k,\ell} - \mathbf{u}_k^{\star}\|_2 = G_{k,\ell} \leq 6(\ell - s_k + 2)m_k^{\ell-s_k+1}$$

which finishes the proof.

## C.2. Proof of Lemma 2

Applying the Davis-Kahan Theorem, if we let $\lambda_{k+1,\ell} = \lambda_{\max}(\boldsymbol{\Sigma}_{k,\ell})$, then for all $k,\ell$ such that $\|\boldsymbol{\Sigma}_k^{\star} - \boldsymbol{\Sigma}_{k,\ell}\|_F < \lambda_k^{\star} - \lambda_{k+1}^{\star}$, we have

$$\|\mathbf{u}_{k,\ell} - \mathbf{u}_k^{\star}\|_2 \leq \frac{\|\boldsymbol{\Sigma}_k^{\star} - \boldsymbol{\Sigma}_{k,\ell}\|_F}{\lambda_k^{\star} - \lambda_{k+1,\ell}}$$

By definition, we have

$$\boldsymbol{\Sigma}_k^\star = \boldsymbol{\Sigma} - \sum_{k'=1}^{k-1} \lambda_{k'}^\star \mathbf{u}_{k'}^\star \mathbf{u}_{k'}^{\star\top}; \quad \boldsymbol{\Sigma}_{k,\ell} = \boldsymbol{\Sigma} - \sum_{k'=1}^{k-1} \left( \mathbf{v}_{k',\ell-1}^\top \boldsymbol{\Sigma} \mathbf{v}_{k',\ell-1} \right) \mathbf{v}_{k',\ell-1} \mathbf{v}_{k',\ell-1}^\top$$

Thus, we can write the difference between the two matrices as

$$\boldsymbol{\Sigma}_k^\star - \boldsymbol{\Sigma}_{k,\ell} = \sum_{k'=1}^{k-1} \left( \lambda_{k'}^\star - \mathbf{v}_{k',\ell-1}^\top \boldsymbol{\Sigma} \mathbf{v}_{k',\ell-1} \right) \mathbf{v}_{k',\ell-1} \mathbf{v}_{k',\ell-1}^\top + \sum_{k'=1}^{k-1} \lambda_{k'}^\star \left( \mathbf{v}_{k',\ell-1} \mathbf{v}_{k',\ell-1}^\top - \mathbf{u}_{k'}^\star \mathbf{u}_{k'}^{\star\top} \right)$$

It is easy to see that for $\mathbf{v}_{k',\ell-1}$ and $\mathbf{v}_{k'}^\star$ with unit norm,

$$\left\| \mathbf{v}_{k',\ell-1} \mathbf{v}_{k',\ell-1}^\top - \mathbf{u}_{k'}^\star \mathbf{u}_{k'}^{\star\top} \right\|_2^2 = 2 - 2 \left\langle \mathbf{v}_{k',\ell-1}, \mathbf{u}_{k'}^\star \right\rangle^2 \le \left\| \mathbf{u}_{k'}^\star - \mathbf{v}_{k',\ell-1} \right\|_2^2$$

Moreover to bound $\left| \lambda_{k'}^\star - \mathbf{v}_{k',\ell-1}^\top \boldsymbol{\Sigma} \mathbf{v}_{k',\ell-1} \right|$, we denote $\boldsymbol{\delta} = \mathbf{v}_{k',\ell-1} - \mathbf{u}_{k'}^\star$, and write

$$\mathbf{v}_{k',\ell-1}^\top \boldsymbol{\Sigma} \mathbf{v}_{k',\ell-1} = \left( \mathbf{u}_{k'}^\star - \boldsymbol{\delta} \right)^\top \boldsymbol{\Sigma} \left( \mathbf{u}_{k'}^\star - \boldsymbol{\delta} \right) = \lambda_{k'}^\star - 2\lambda_{k'}^\star \boldsymbol{\delta}^\top \mathbf{u}_{k'}^\star + \boldsymbol{\delta}^\top \boldsymbol{\Sigma} \boldsymbol{\delta}$$

Therefore, we have

$$\left| \lambda_{k'}^\star - \mathbf{v}_{k',\ell-1}^\top \boldsymbol{\Sigma} \mathbf{v}_{k',\ell-1} \right| = \left| -2\lambda_{k'} \boldsymbol{\delta}^\top \mathbf{u}_{k'}^\star + \boldsymbol{\delta}^\top \boldsymbol{\Sigma} \boldsymbol{\delta} \right| \le 2\lambda_{k'}^\star \left\| \mathbf{v}_{k',\ell-1} - \mathbf{u}_{k'}^\star \right\|_2 + \lambda_1^\star \left\| \mathbf{v}_{k',\ell-1} - \mathbf{u}_{k'}^\star \right\|_2^2$$

This gives

$$\left\| \boldsymbol{\Sigma}_k^\star - \boldsymbol{\Sigma}_{k,\ell} \right\|_F \le \sum_{k'=1}^{k-1} \left( 3\lambda_{k'}^\star \left\| \mathbf{u}_{k'}^\star - \mathbf{v}_{k',\ell-1} \right\|_2 + \lambda_1^\star \left\| \mathbf{u}_{k'}^\star - \mathbf{v}_{k',\ell-1} \right\|_2^2 \right)$$

We then need to assume that, for some $c_0 > 1$,

$$\sum_{k'=1}^{k-1} \lambda_{k'}^\star \left\| \mathbf{u}_{k'}^\star - \mathbf{v}_{k',\ell-1} \right\|_2 \le \frac{c_0 - 1}{4c_0} \left( \lambda_k^\star - \lambda_{k+1}^\star \right)$$

In this scenario, we can conclude that $\left\| \mathbf{u}_{k'}^\star - \mathbf{v}_{k',\ell-1} \right\|_2 \le \lambda_k^\star$. Combined with the condition that $\lambda_1^\star = 1$, we have

$$\left\| \boldsymbol{\Sigma}_k^\star - \boldsymbol{\Sigma}_{k,\ell} \right\|_F \le 4 \sum_{k'=1}^{k-1} \lambda_{k'}^\star \left\| \mathbf{u}_{k'}^\star - \mathbf{v}_{k',\ell-1} \right\|_2 \le \frac{c_0 - 1}{c_0} \left( \lambda_k^\star - \lambda_{k+1}^\star \right)$$

Moreover, we have

$$\left\| \mathbf{u}_{k,\ell} - \mathbf{u}_k^\star \right\|_2 \le \frac{4}{\lambda_k^\star - \lambda_{k+1,\ell}} \sum_{k'=1}^{k-1} \lambda_{k'}^\star \left\| \mathbf{u}_{k'}^\star - \mathbf{v}_{k',\ell-1} \right\|_2 \le \frac{4c_0}{\lambda_k^\star - \lambda_{k+1}^\star} \sum_{k'=1}^{k-1} \lambda_{k'}^\star \left\| \mathbf{u}_{k'}^\star - \mathbf{v}_{k',\ell-1} \right\|_2$$

where the last inequality follows from $\lambda_k^\star - \lambda_{k+1,\ell} \ge \lambda_k^\star - \lambda_{k+1}^\star - \left\| \boldsymbol{\Sigma}_k^\star - \boldsymbol{\Sigma}_{k,\ell} \right\|_F \ge \frac{1}{c_0} \left( \lambda_k^\star - \lambda_{k+1}^\star \right)$

## C.3. Proof of Lemma 3

To start, we will need to prove an auxiliary lemma

**Lemma 5.** *Let the sequence $\{\hat{B}_{k,\ell}\}_{\ell=s_{k-1}+1}^\infty$ be defined as*

$$\hat{B}_{k,\ell} = \min \left\{ 2, m_{k-1}^{\ell-\hat{s}_k} (\ell - \hat{s}_k + 1) \hat{B}_{k,\hat{s}_k} \right\}$$

*with some $\hat{B}_{k,\hat{s}_k} \le 2$ and $m_{k-1} \in (0,1)$. Then for all $s$ that satisfies $m_{k-1}^{s-\hat{s}_k} \le \frac{1}{s-\hat{s}_k+1}$ and $s \ge \frac{km_{k-1}}{1-m_{k-1}} + \hat{s}_k$, we have that*

$$\hat{B}_{k,\ell} \le \left( \frac{1}{k} + \frac{k-1}{k} m_{k-1} \right)^{\ell-s} \hat{B}_{k,s}$$

*Proof.* To start, by definition, we can write $\hat{B}_{k,s}$ as

$$\hat{B}_{k,s} = \min\left\{2, m_{k-1}^{s-\hat{s}_k}(s-\hat{s}_k+1)\hat{B}_{k,\hat{s}_k}\right\}$$

Since $\hat{B}_{k,\hat{s}_k} \leq 2$, we have

$$m_{k-1}^{s-\hat{s}_k}(s-\hat{s}_k+1)\hat{B}_{k,\hat{s}_k} \leq 2m_{k-1}^{s-\hat{s}_k}(s-\hat{s}_k+1) \leq 2$$

where the last inequality follows from the condition $m_{k-1}^{s-\hat{s}_k} \leq \frac{1}{s-\hat{s}_k+1}$. Therefore, we can write $\hat{B}_{k,s}$ as

$$\hat{B}_{k,s} = m_{k-1}^{s-\hat{s}_k}(s-\hat{s}_k+1)\hat{B}_{k,\hat{s}_k}$$

Recall that for any $\ell \geq s$ we have

$$\hat{B}_{k,\ell} = \min\left\{2, m_{k-1}^{\ell-\hat{s}_k}(\ell-\hat{s}_k+1)\hat{B}_{k,\hat{s}_k}\right\}$$

Plugging in $\hat{B}_{k,\hat{s}_k} = \left(m_{k-1}^{s-\hat{s}_k}(s-\hat{s}_k+1)\right)^{-1}\hat{B}_{k,s}$ we have

$$\hat{B}_{k,\ell} \leq m_{k-1}^{\ell-s} \cdot \frac{\ell-\hat{s}_k+1}{s-\hat{s}_k+1} \cdot \hat{B}_{k,s}$$

$$\leq m_k^{\ell-s}\left(\prod_{\ell'=s+1}^{\ell} \frac{\ell'-\hat{s}_k+1}{\ell'-\hat{s}_k}\right) \cdot \hat{B}_{k,s}$$

$$\leq m_k^{\ell-s}\left(\frac{s-\hat{s}_k+1}{s-\hat{s}_k}\right)^{\ell-s}\hat{B}_{k,s}$$

$$= \left(m_k \cdot \frac{s-\hat{s}_k+1}{s-\hat{s}_k}\right)^{\ell-s}\hat{B}_{k,s}$$

$$\leq \left(\frac{1}{k}+\frac{k-1}{k}m_k\right)^{\ell-s}\hat{B}_{k,s}$$

where the last inequality is because $s \geq \frac{km_k}{1-m_k}+\hat{s}_k$ implies that

$$m_k \cdot \frac{s-\hat{s}_k+1}{s-\hat{s}_k} \leq m_k \cdot \frac{\frac{km_k}{1-m_k}+1}{\frac{km_k}{1-m_k}} = \frac{(k-1)m_k+1}{k} = \frac{1}{k}+\frac{k-1}{k}m_k$$

This completes the proof. $\qquad\square$

We will use induction on $k$ to prove the lemma.

**Base Case:** $k = 1$. In this case, by the definition of $\hat{B}_{1,\hat{s}_1}$, we have $\hat{B}_{1,s_1} = 0$. Moreover, by the definition of $\hat{B}_{1,\ell}$ for $\ell \geq s_1$, we have $\hat{B}_{1,\ell} = m_0^{\ell-\hat{s}_k}(\ell-\hat{s}_k+1)\hat{B}_{k,\hat{s}_k} = 0$. Lastly, by the definition of $B_{1,\ell}$, we have $B_{1,\ell} = 0$. Therefore, we must have that $\hat{B}_{1,\ell} = 0 = B_{1,\ell}$ for all $\ell \geq \hat{s}_1$. This shows Condition #1. Using $\hat{B}_{1,\ell} = 0 = B_{1,\ell}$, we can derive that $\hat{G}_{1,\ell} = \mathcal{F}_1^{\ell-s_1+1}D_{1,s_1-1}$, and $G_{1,\ell} = \mathcal{F}_1^{\ell-s_1+1}D_{1,s_1-1}$. This implies that $G_{1,\ell} \leq \hat{G}_{1,\ell}$, and shows Condition #2. Thus, we have shown that the case $k = 1$ holds.

**Inductive Step.** Now, we assume that for all $\hat{k} \leq k$, the following holds

1. $\hat{B}_{\hat{k},\ell} \geq B_{\hat{k},\ell}$ for all $\ell \geq \hat{s}_{\hat{k}}$

2. $\hat{G}_{\hat{k},\ell} \geq G_{\hat{k},\ell}$ for all $\ell \geq s_{\hat{k}}-1$

We wish to show that the above three conditions hold for $\hat{k} = k+1$. We start by showing Condition #1 for $\hat{k} = k+1$. By Condition #2 in the inductive hypothesis, we have that $\hat{G}_{\hat{k},\ell} \geq G_{\hat{k},\ell}$ for all

$\ell \geq s_{\hat{k}} - 1$. Since $\hat{s}_{k+1} \geq s_{\hat{k}}$ for all $\hat{k} \leq k$, we have that $\hat{G}_{\hat{k},\ell} \geq G_{\hat{k},\ell}$ for all $\ell \geq \hat{s}_{k+1} - 1$. Therefore, in the case of $\ell = \hat{s}_{k+1}$

$$B_{k+1,\ell} \leq C_{k+1} \sum_{k'=1}^{k} \lambda_{k'}^\star G_{k',\ell-1} \leq C_{k+1} \sum_{k'=1}^{k} \lambda_{k'}^\star \hat{G}_{k',\ell-1} = \hat{B}_{k,\ell}$$

Next, we show that $\hat{B}_{k+1,\ell} \geq B_{k+1,\ell}$ for all $\ell \geq \hat{s}_{k+1}$. If $\hat{B}_{k+1,\ell} \geq 2$, then we directly have $\hat{B}_{k+1,\ell} \geq B_{k+1,\ell}$ since $B_{k+1,\ell} \leq 2$. Otherwise, suppose $\hat{B}_{k+1,\ell} \leq 2$. Since $\hat{s}_{k+1} \geq s_{k'}$ for all $k' \leq k$, by the definition of $\hat{G}_{k,\ell}$, we have

$$\hat{G}_{k',\hat{s}_{k+1}-1} = m_{k'}^{\hat{s}_{k+1}-s_{k'}} (\hat{s}_{k+1} - s_{k'} + 1) \hat{G}_{k',s_{k'}-1}$$

Based on the definition of $\hat{B}_{k+1,s_k}$, and since $m_k \geq m_{k'}$ for all $k \geq k$, we have that

$$\begin{aligned}
\hat{B}_{k+1,\ell} &= m_k^{\ell-\hat{s}_{k+1}} (\ell - \hat{s}_{k+1} + 1) \hat{B}_{k+1,\hat{s}_{k+1}} \\
&= m_k^{\ell-\hat{s}_{k+1}} (\ell - \hat{s}_{k+1} + 1) C_{k+1} \sum_{k'=1}^{k} \lambda_{k'}^\star \hat{G}_{k',\hat{s}_{k+1}-1} \\
&\geq C_{k+1} \sum_{k'=1}^{k} \lambda_{k'}^\star m_{k'}^{\ell-\hat{s}_{k+1}} (\ell - \hat{s}_{k+1} + 1) \hat{G}_{k',\hat{s}_{k+1}-1} \\
&\geq C_{k+1} \sum_{k'=1}^{k} \lambda_{k'}^\star m_{k'}^{\ell-s_{k'}} (\ell - \hat{s}_{k+1} + 1) (s_{k+1} - s_{k'} + 1) \hat{G}_{k',s_{k'}-1} \\
&\geq C_{k+1} \sum_{k'=1}^{k} \lambda_{k'}^\star m_{k'}^{\ell-s_{k'}} (\ell - s_{k'} + 1) \hat{G}_{k',s_{k'}-1} \\
&= C_{k+1} \sum_{k'=1}^{k} \lambda_{k'}^\star \hat{G}_{k',\ell-1}
\end{aligned}$$

where the third to the last inequality is due to $(\ell - \hat{s}_{k+1} + 1) + (s_{k+1} - s_{k'} + 1) - 1 = \ell - s_{k'} + 1$, and for all $a \geq 1, b \geq 1$, we will have $ab \geq a + b - 1$. By the inductive hypothesis, we have that $\hat{G}_{k',\ell} \geq G_{k',\ell}$ for all $\ell \geq \hat{s}_{k+1} \geq s_{k'} - 1$. Therefore, it must hold that

$$\hat{B}_{k+1,\ell} \geq C_{k+1} \sum_{k'=1}^{k} \lambda_{k'}^\star G_{k',\ell-1} \geq B_{k+1,\ell}$$

This proves Condition #1 for $\hat{k} = k + 1$. Next, we will prove Condition #2 for $\hat{k} = k + 1$. To start, when $\ell = s_{k+1} - 1$, we have

$$\hat{G}_{k+1,\ell} = D_{k+1,s_{k+1}-1} + \hat{B}_{k+1,s_{k+1}-1} + \hat{B}_{k+1,s_{k+1}-2}$$

while by (18) we have

$$G_{k+1,\ell} \leq D_{k+1,s_{k+1}-1} + B_{k+1,s_{k+1}-1}$$

Since $\hat{B}_{k+1,s_{k+1}-2} \geq 0$ and $\hat{B}_{k+1,s_{k+1}-1} \geq B_{k+1,s_{k+1}-1}$ as proved above for $s_{k+1} \geq \hat{s}_{k+1} - 2$, we must have that $\hat{G}_{k+1,\ell} \geq G_{k+1,\ell}$ when $\ell = s_{k+1} - 1$. Next, we show that $\hat{G}_{k+1,\ell} \geq G_{k+1,\ell}$ when $\ell > s_{k+1} - 1$. To start,

$$\begin{aligned}
G_{k+1,\ell} &\leq \mathcal{F}_{k+1}^{\ell-s_{k+1}+1} D_{k+1,s_{k+1}-1} + \sum_{\ell'=s_{k+1}-1}^{\ell-1} \mathcal{F}_{k+1}^{\ell-\ell'} (B_{k+1,\ell'} + B_{k+1,\ell'-1}) + B_{k+1,\ell} \\
&\leq \mathcal{F}_{k+1}^{\ell-s_{k+1}+1} D_{k+1,s_{k+1}-1} + \sum_{\ell'=s_{k+1}-1}^{\ell} \mathcal{F}_{k+1}^{\ell-\ell'} (B_{k+1,\ell'} + B_{k+1,\ell'-1}) \\
&\leq \mathcal{F}_{k+1}^{\ell-s_{k+1}+1} D_{k+1,s_{k+1}-1} + \sum_{\ell'=s_{k+1}-1}^{\ell} \mathcal{F}_{k+1}^{\ell-\ell'} \left( \hat{B}_{k+1,\ell'} + \hat{B}_{k+1,\ell'-1} \right)
\end{aligned}$$

By definition of $\hat{B}_{k+1,\ell}$, invoking Lemma 5 with $s = s_{k+1} - 2$ and $s = s_{k+1} - 1$, we have that, as long as $s_{k+1}$ satisfies $m_k^{s_{k+1}-\hat{s}_{k+1}-2} \leq \frac{1}{s_{k+1}-\hat{s}_{k+1}-1}$ and $s_{k+1} \geq \frac{(k+1)m_k}{1-m_k} + \hat{s}_{k+1} + 2$, it holds that

$$\hat{B}_{k+1,\ell} \leq \gamma_k^{\ell-s_{k+1}+2}\hat{B}_{k+1,s_{k+1}-2}$$

$$\hat{B}_{k+1,\ell} \leq \gamma_k^{\ell-s_{k+1}+1}\hat{B}_{k+1,s_{k+1}-1}$$

for all $\ell \geq s_k$. Therefore

$$G_{k+1,\ell} \leq \mathcal{F}_{k+1}^{\ell-s_{k+1}+1}D_{k+1,s_{k+1}-1} + \sum_{\ell'=s_{k+1}-1}^{\ell} \mathcal{F}_{k+1}^{\ell-\ell'}\left(\hat{B}_{k+1,\ell'} + \hat{B}_{k+1,\ell'-1}\right)$$

$$= \mathcal{F}_{k+1}^{\ell-s_{k+1}+1}D_{k+1,s_{k+1}-1} + \sum_{\ell'=s_{k+1}-1}^{\ell} \mathcal{F}_{k+1}^{\ell-\ell'}\gamma_k^{\ell'-s_{k+1}+1}\left(\hat{B}_{k+1,s_{k+1}-1} + \hat{B}_{k+1,s_{k+1}-2}\right)$$

$$\leq \max\{\mathcal{F}_{k+1},\gamma_k\}^{\ell-s_{k+1}+1}\left(D_{k+1,s_{k+1}-1} + (\ell - s_{k+1} + 1)\left(\hat{B}_{k+1,s_{k+1}-1} + \hat{B}_{k+1,s_{k+1}-2}\right)\right)$$

$$\leq \max\{\mathcal{F}_{k+1},\gamma_k\}^{\ell-s_{k+1}+1}(\ell - s_{k+1} + 1)\left(D_{k+1,s_{k+1}-1} + \hat{B}_{k+1,s_{k+1}-1} + \hat{B}_{k+1,s_{k+1}-2}\right)$$

$$= \hat{G}_{k+1,\ell}$$

where in the last equality we use $m_{k+1} = \max\{\mathcal{F}_{k+1},\gamma_k\}$ and

$$\hat{G}_{k+1,s_{k+1}-1} = D_{k+1,s_{k+1}-1} + \hat{B}_{k+1,s_{k+1}-1} + \hat{B}_{k+1,s_{k+1}-2}$$

This proves Condition #2 under $\ell > s_{k+1} - 1$, which finishes the induction step and completes the proof.

### C.4. Proof of Lemma 4

First, we prove the case $\epsilon \geq -\frac{m}{e\log m}$. Notice that the function $g(x)$ achieves global maximum at $x = \frac{1}{\log 1/m} - 1$ with value $\frac{1}{\log 1/m}m^{\frac{1}{\log 1/m}-1}$. Moreover, notice that

$$\frac{1}{\log 1/m}m^{\frac{1}{\log 1/m}-1} = -\frac{m^{\frac{1}{-\log m}}}{m\log m} = -\frac{e^{-\frac{1}{\log m}\cdot\log m}}{m\log m} = -\frac{1}{em\log m} \leq \epsilon$$

Therefore, for all $x \geq 0$ we would have $g(x) \leq \epsilon$. Next, we consider the case $\epsilon \leq -\frac{1}{em\log m}$. In this case, $x \geq \frac{1}{\log m}W_{-1}\left(\epsilon m\log m\right) - 1$ implies that

$$(x + 1)\log m \leq W_{-1}\left(\epsilon m\log m\right)$$

By the monotonicity of $W_{-1}$, we have

$$(x + 1)\log m \cdot e^{(x+1)\log m} \geq \epsilon m\log m$$

which gives $(x + 1)e^{x\log m} \leq \epsilon$. Thus, we have $g(x) = (x + 1)m^x \leq \epsilon$.

## D. Ablation studies.

We conducted additional ablation studies for the parallel deflation algorithms, with the results presented in Figure 4. In Figure 4a, we conduct additional experiments comparing how different choices of the number of local updates $T$ contribute to the convergence speed. In particular, since in each communication round, the local updates of all workers are done in parallel, $T \times \ell$ would represent the total time elapsed under the ideal scenario of no communication cost. We could see from Figure 4a that a smaller $T$ results in a faster convergence speed. Remarkably, since we choose $T \times L = 1200$ and aim at recovering 30 eigenvectors, the case where $T = 40$ corresponds to $L = 30$, demonstrating the convergence behavior of the sequential deflation algorithm. Figure 4a thus supports that introducing additional parallelism into the deflation algorithm indeed speeds up the computation process. On the other hand, Figure 4b considers the case where the communication cost is the major burden. In this case, we can run the parallel deflation algorithm with a larger number of local updates, hoping to make more progress within one communication round. Indeed, Figure 4b shows that a large number of local updates result in a faster convergence within a fixed number of communication rounds on larger datasets.

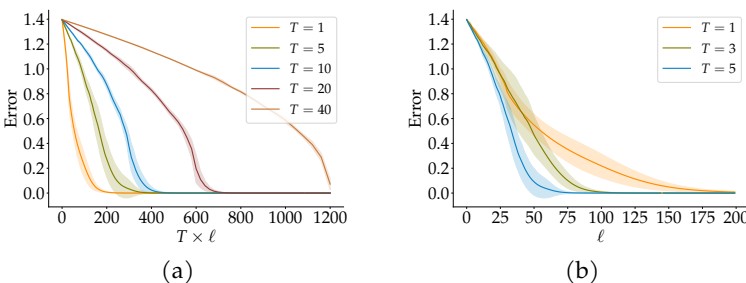

Figure 4: Ablation study of the parallel deflation algorithm. (a) shows the benefit of the run-time by increasing the parallelism. (b) shows the benefit of decreasing the communication cost by increasing the number of local iterations.

# E. Stochastic Parallel Deflation Algorithm

In this section, we provide the explicit form of the stochastic version of the parallel deflation algorithm as discussed in Section 3. Notice that in this algorithm we choose Hebb's rule as the $\text{Top} - 1$ subroutine for the convenience of a clearer presentation. However, any subroutine that use $\mathbf{\Sigma}_{k,\ell}$ only for a matrix-vector multiplication can enjoy a similar efficient implementation.

---

**Algorithm 2** Stochastic Parallel Deflation with Hebb's Rule

---

**Require:** Batch of data in the $(\ell,t)$th iteration $\hat{\mathbf{Y}}_{\ell,t}$; # of eigenvectors (workers) $K$; # of iterations $T$; global communication rounds $L \geq K$, step size $\eta$.
**Ensure:** Approximate eigenvectors $\{\mathbf{v}_k\}_{k=1}^K$.
1: **for** $k = 1, \ldots, K$ **do**
2:      Randomly initialize $\hat{\mathbf{v}}_{k,\texttt{init}}$ with unit norm;
3: **end for**
4: **for** $\ell = 1, \ldots, L$ **do**
5:      **for** $k = 1, \ldots, K$ **do**
6:          **if** $k \leq \ell$ **then**
7:              Receive $\mathbf{v}_{1,\ell-1}, \ldots, \mathbf{v}_{k-1,\ell-1}$
8:              $\mathbf{v}_{k,\ell,0} := \mathbf{v}_{k,\ell-1}$;
9:              **for** $t = 1, \ldots, T$ **do**
10:                  $\hat{\lambda}_{k',\ell,t} = \|\hat{\mathbf{Y}}_{\ell,t}\mathbf{v}_{k',\ell-1}\|_2^2 \ \ \forall k' \in [k-1]$;
11:                  $\mathbf{g}_{k,\ell,t} = \hat{\mathbf{Y}}_{k,\ell,t}^\top \hat{\mathbf{Y}}_{k,\ell,t}\mathbf{v}_{k,\ell,t-1} - \sum_{k'=1}^{k-1} \hat{\lambda}_{k',\ell,t} \left(\mathbf{v}_{k',\ell-1}^\top \mathbf{v}_{k,\ell,t-1}\right) \cdot \mathbf{v}_{k',\ell-1}$
12:                  $\mathbf{v}_{k,\ell,t} := (\mathbf{v}_{k,\ell,t-1} - \eta\mathbf{g}_{k,t,\ell})/\|\mathbf{v}_{k,\ell,t-1} - \eta\mathbf{g}_{k,t,\ell}\|_2$
13:              **end for**
14:              Broadcast $\mathbf{v}_{k,\ell} := \mathbf{v}_{k,\ell,T}$
15:          **else**
16:              $\mathbf{v}_{k,\ell} := \hat{\mathbf{v}}_{k,\texttt{init}}$;
17:          **end if**
18:      **end for**
19: **end for**
20: **return** $\{\mathbf{v}_{k,L}\}_{k=1}^K$

---

# F. Baseline Algorithms

We provide the generalization of the EigenGame-$\alpha$ [33] and EigenGame-$\mu$ [34] algorithms with multiple iterations of local updates $T \geq 1$ in Algorithm 3 and Algorithm 4. In particular, it should be noted that EigenGame-$\alpha$ and EigenGame-$\mu$ use covariance matrices computed on subsets of the data in each iteration, where in our case we assume that the covariance matrix is computed on the whole dataset before the algorithm runs. Moreover, if we set $T = 1$ in both Algorithm 3 and Algorithm 4, then we recover the original EigenGame-$\alpha$ and EigenGame-$\mu$ algorithms.

---

**Algorithm 3** EigenGame-$\alpha$

---

**Require:** $\boldsymbol{\Sigma} \in \mathbb{R}^{d \times d}$; # of eigenvectors (workers) $K$; # of iterations $T$; global communication rounds $L \geq K$, step size $\eta$.

**Ensure:** Approximate eigenvectors $\{\mathbf{v}_k\}_{k=1}^{K}$.

1: **for** $k = 1, \ldots, K$ **do**
2:      Randomly initialize $\hat{\mathbf{v}}_{k,\texttt{init}}$ with unit norm;
3: **end for**
4: **for** $\ell = 1, \ldots, L$ **do**
5:      **for** $k = 1, \ldots, K$ **do**
6:          **if** $k \leq \ell$ **then**
7:              Receive $\mathbf{v}_{1,\ell-1}, \ldots, \mathbf{v}_{k-1,\ell-1}$
8:              $\mathbf{v}_{k,\ell,0} := \mathbf{v}_{k,\ell-1}$;
9:              **for** $t = 1, \ldots, T$ **do**
10:                  $\mathbf{g}_{k,\ell,t} := \boldsymbol{\Sigma}\mathbf{v}_{k,\ell,t-1} - \sum_{k'=1}^{k-1} \frac{\mathbf{v}_{k',\ell-1}^{\top}\boldsymbol{\Sigma}\mathbf{v}_{k,\ell,t-1}}{\mathbf{v}_{k',\ell-1}^{\top}\boldsymbol{\Sigma}\mathbf{v}_{k',\ell-1}} \cdot \boldsymbol{\Sigma}\mathbf{v}_{k',\ell-1}$;
11:                  $\mathbf{v}_{k,\ell,t} := \left(\mathbf{v}_{k,\ell,t-1} - \eta\mathbf{g}_{k,t,\ell}\right)/\left\|\mathbf{v}_{k,\ell,t-1} - \eta\mathbf{g}_{k,t,\ell}\right\|_2$
12:              **end for**
13:              Broadcast $\mathbf{v}_{k,\ell} := \mathbf{v}_{k,\ell,T}$
14:          **else**
15:              $\mathbf{v}_{k,\ell} := \hat{\mathbf{v}}_{k,\texttt{init}}$;
16:          **end if**
17:      **end for**
18: **end for**
19: **return** $\left\{\mathbf{v}_{k,L}\right\}_{k=1}^{K}$

---

**Algorithm 4** EigenGame-$\mu$

---

**Require:** $\boldsymbol{\Sigma} \in \mathbb{R}^{d \times d}$; # of eigenvectors (workers) $K$; # of iterations $T$; global communication rounds $L \geq K$, step size $\eta$.

**Ensure:** Approximate eigenvectors $\{\mathbf{v}_k\}_{k=1}^{K}$.

1: **for** $k = 1, \ldots, K$ **do**
2:      Randomly initialize $\hat{\mathbf{v}}_{k,\texttt{init}}$ with unit norm;
3: **end for**
4: **for** $\ell = 1, \ldots, L$ **do**
5:      **for** $k = 1, \ldots, K$ **do**
6:          **if** $k \leq \ell$ **then**
7:              Receive $\mathbf{v}_{1,\ell-1}, \ldots, \mathbf{v}_{k-1,\ell-1}$
8:              $\mathbf{v}_{k,\ell,0} := \mathbf{v}_{k,\ell-1}$;
9:              **for** $t = 1, \ldots, T$ **do**
10:                  $\mathbf{g}_{k,\ell,t} := \boldsymbol{\Sigma}\mathbf{v}_{k,\ell,t-1} - \sum_{k'=1}^{k-1} \mathbf{v}_{k',\ell-1}^{\top}\boldsymbol{\Sigma}\mathbf{v}_{k,\ell,t-1} \cdot \mathbf{v}_{k',\ell-1}$;
11:                  $\mathbf{v}_{k,\ell,t} := \left(\mathbf{v}_{k,\ell,t-1} - \eta\mathbf{g}_{k,t,\ell}\right)/\left\|\mathbf{v}_{k,\ell,t-1} - \eta\mathbf{g}_{k,t,\ell}\right\|_2$
12:              **end for**
13:              Broadcast $\mathbf{v}_{k,\ell} := \mathbf{v}_{k,\ell,T}$
14:          **else**
15:              $\mathbf{v}_{k,\ell} := \hat{\mathbf{v}}_{k,\texttt{init}}$;
16:          **end if**
17:      **end for**
18: **end for**
19: **return** $\left\{\mathbf{v}_{k,L}\right\}_{k=1}^{K}$

---

