# OpenReview forum: "Provable Model-Parallel Distributed Principal Component Analysis with Parallel Deflation"
_CPAL.cc/2025/Proceedings_Track — CPAL 2025 (Proceedings Track) Poster_

### Official Review · Reviewer_AA6N · 2025-01-06
**Good theoretical and empirical results**

**Rating:** 6
**Confidence:** 2

**Review:**

This paper presents and analyses an algorithm for distributed PCA by computing different eigenvalues/vectors in parallel.

Pros:
- Theoretical contribution while also showing efficacy in experiments

Cons/comments/questions:
1) Lines in Figures 2 and 3 were a bit hard to differentiate. Please consider adding markers or some other visual cue.
2) How does your method perform with varying levels of noise in the data?
3) Why does EigGame-$\alpha$, T=1 converge much slower than the other T=1 methods in Figure 2?
4) Do the analyses hold with repeat eigenvalues (or in the extreme case, all identical eigenvalues)?

---

### Official Review · Reviewer_ArM6 · 2025-01-06
**Review of "Provable Model-Parallel Distributed Principal Component Analysis with Parallel Deflation"**

**Rating:** 7
**Confidence:** 4

**Review:**

## Summary

This paper advances **model-parallel distributed PCA** by building upon the collaborative computation framework introduced in previous works (EigenGame [33, 34]). The key innovation is a **two-stage parallel deflation algorithm** (Section 3, Algorithm 1) designed to alleviate the strict sequential dependencies typical of classical deflation-based methods:

1. ***Stage 1 (Initial Estimation)*** – Executed in a sequential manner where each worker obtains a “rough” estimate of its assigned eigenvector, requiring relatively few iterations.
2. ***Stage 2 (Iterative Refinement)*** – Conducted in parallel, where each worker refines its eigenvector by leveraging updated information from the other workers.

By formalizing how parallel computations impact convergence rates (Section 4), the authors offer a new theoretical benchmark for distributed PCA algorithms. They show that their parallel deflation scheme converges under suitable conditions and provide insights into how synchronization and parameter updates should be managed to ensure orthogonality among eigenvectors. In Section 5, the authors validate their approach empirically, demonstrating that the **proposed P-Deflation** algorithm performs competitively with EigenGame and scales more effectively in a distributed setting.

---

## Evaluation

Overall, this paper presents **sound theoretical results** and provides **clear exposition** of the proposed algorithm. The two-stage approach represents a practical refinement to traditional deflation, offering a path toward **model-parallel** computation—an aspect that is often overshadowed by data-parallel PCA methods. Below, we discuss the paper’s strengths and identify areas for further improvement.

### Pros

1. **Rigorous and Novel Theoretical Framework**
   - The paper systematically addresses the **interaction between parallel computation and convergence** in PCA, providing proofs that strengthen the foundation of distributed deflation methods.

2. **Parallel Deflation Approach**
   - The two-stage design usefully relaxes the **strict sequential dependencies** in traditional deflation. Stage 1 offers coarse but rapid initialization, and Stage 2 refines estimates in parallel, improving scalability.

3. **Clear and Well-Structured Presentation**
   - The manuscript is **well-organized**, with each section building on the previous one. The step-by-step description of Algorithm 1, followed by theoretical results in Section 4 and empirical validation in Section 5, conveys a coherent narrative.

4. **Promising Empirical Validation**
   - Numerical experiments compare **P-Deflation** with **EigenGame**, demonstrating competitive performance. The results suggest that the proposed method can be effectively applied to moderate-scale problems.

5. **Potential for Broader Impact**
   - The work highlights that **model-parallel PCA** is not merely a theoretical curiosity: it may be relevant to more general ML scenarios involving partitioned parameter sets or extremely high-dimensional data where each component vector is expensive to compute or store on a single device.

### Cons

1. **Limited Motivation for Model-Parallel Approaches**
   - While the paper improves upon EigenGame, the **practical need** for model-parallel PCA remains under-discussed. The authors could clarify whether this is a stylized setting for broader model-parallel ML tasks, or whether PCA itself necessitates distributed parameter storage when the dimension is huge.

2. **Lack of Wider Experimental Benchmarks**
   - Although the paper demonstrates viability on synthetic examples (and references real-world datasets to some extent), incorporating **more diverse or larger-scale** tasks (especially beyond standard image sets) would strengthen the paper’s empirical scope.

3. **Minor Issues & Suggestions**
   - In Definitions 1 & 2, the dependence on the argument **_v_** (presumably the initialization) is unclear in the current statements. Making these definitions more explicit would reduce ambiguity for readers following the convergence arguments.
   - The plots in Figures 2 and 3 are somewhat difficult to read, with multiple overlapping curves. Emphasizing the P-Deflation lines (e.g., by using thicker lines, distinct color palettes, or markers) would improve readability.
   - The paper could benefit from a more detailed discussion of communication overhead in large-scale or high-latency environments, where repeated synchronization among workers might limit real-world speedups.

---

### Official Review · Reviewer_bmWx · 2025-01-07
**This paper introduces P-Deflation, a novel distributed Principal Component Analysis (PCA) framework. The authors provide rigorous theoretical analysis and experimental evidence to validate the convergence and performance of their method.**

**Rating:** 6
**Confidence:** 2

**Review:**

### Quality

The quality of the paper is strong. It presents thorough theoretical foundations and includes detailed experimental comparisons to support the proposed framework.

### Clarity

The paper is well-organized, and the concepts are presented clearly.

### Originality

The paper is a novel.

### Significance

The significance of this work is OK. While the method is innovative and supported by solid theory, its practical performance, particularly in stochastic settings, shows limited improvement over existing baselines. This may reduce its impact in real-world applications.

### Pros

The paper has solid theoretical support for the proposed method. They also demonstrated performance gains over the baseline under certain experimental conditions.

### Cons
1. In line 261, the paper has $s_1 = 1$, which means the first eigenvectors’ error recovery enters the linear convergence phase in the first communication round. Is this an assumption or a conclusion from the previous papers?
2. In line 270, when $m_0 = 1$, $m_1 = max ( F_k, 1/k + (k-1)m_{k-1}/k ) = 1$ and all other $m_k = 1$, how to get $m_k < 1$?
3. The color in Figure 2 and Figure 3 is very hard to distinguish between different methods.
4. The baseline comparisons are limited. Including additional efficient PCA methods would better highlight the strengths and weaknesses of P-Deflation, like generalized power iteration for multiple eigenvectors.
5. Could the author give some intuitions as to why their method outperforms EigenGame? Either in terms of theoretical converge rate or high-level intuition.

---

### Meta-Review · Area_Chair_nJSf · 2025-02-02

**Recommendation:** Accept (Poster)
**Confidence:** 4

**Metareview:**

All three reviewers seem to find that the paper offers solid theoretical insights into distributed PCA with parallel deflation and reports promising empirical results that are competitive with existing methods like EigenGame. The paper’s theoretical contributions and clear exposition are noted as strengths. However, reviewers note relatively limited baseline comparisons, minor presentation issues with figures, and requests for additional experiments (e.g., varying noise levels). Overall, the contributions are deemed to be of sufficient novelty and rigor, and the consensus is to accept.

---

### Decision · Program_Chairs · 2025-02-11

Accept (Poster)